

# Holocene sea-ice dynamics in Petermann Fjord

Henrieka Detlef[1,2], Brendan Reilly[3], Anne Jennings[4], Mads Mørk Jensen[5], Matt O'Regan[6], Marianne Glasius[5], Jesper Olsen[2,7,8], Martin Jakobsson[6], Christof Pearce[1,2]

[1]Department of Geoscience, Aarhus University, Høegh-Guldbergs Gade 2, 8000 Aarhus C, Denmark
[2]Arctic Research Centre, Aarhus University, Ny Munkegade 114, 8000 Aarhus C, Denmark
[3]Scripps Institution of Oceanography, University of California San Diego, La Jolla, California, USA
[4]Institute of Arctic and Alpine Research, University of Colorado, Boulder, CO 80309-0450, USA
[5]Department of Chemistry, Aarhus University, Langelandsgade 140, 8000 Aarhus C, Denmark
[6]Department of Geological Sciences, Stockholm University, 106 91 Stockholm, Sweden
[7]Department of Physics and Astronomy, Aarhus University, Ny Munkegade 120, 8000 Aarhus C, Denmark
[8]School of Culture and Society – Centre for Urban Network Evolutions, Moesgård Alle 20, 8270 Højbjerg, Denmark

*Correspondence to*: Henrieka Detlef (henrieka.detlef@geo.au.dk)

**Abstract.** *The Petermann 2015 Expedition* to Petermann Fjord and adjacent Hall Basin recovered a transect of cores from Nares Strait to under the 48 km long ice tongue of Petermann glacier, offering a unique opportunity to study ice-ocean-sea ice interactions at the interface of these realms. First results suggest that no ice tongue existed in Petermann Fjord for large parts of the Holocene, raising the question of the role of the ocean and the marine cryosphere in the collapse and re-establishment of the ice tongue. Here we use a multi-proxy approach (sea-ice related biomarkers, total organic carbon and its carbon isotopic composition, and benthic and planktonic foraminiferal abundances) to explore Holocene sea-ice dynamics at OD1507-03TC-41GC-03PC in outer Petermann Fjord. Our results are in line with a tight coupling of the marine and terrestrial cryosphere in this region and, in connection with other regional sea-ice reconstructions, give insights into the Holocene evolution of ice arches and associated landfast ice in Nares Strait.

The late stages of the regional Holocene Thermal Maximum (5,500-6,900 cal yrs BP) were marked by reduced seasonal sea-ice concentrations in Nares Strait and the lack of ice arch formation. This was followed by a transitional period towards neoglacial cooling from 3,500-5,500 cal yrs BP, where a southern ice arch might have formed, but an early seasonal break-up and late formation likely caused a prolonged open water season and enhanced pelagic productivity in Nares Strait. Between 1,400 cal yrs BP and 3,500 cal yrs BP, regional records suggest the formation of a stable northern ice arch only, with a short period from 2,100-2,500 cal yrs BP where a southern ice arch might have persisted in response to atmospheric cooling spikes. A stable southern ice arch, or even double arching, is also inferred for the period after 1,400 cal yrs BP. Thus, both the inception of a small Petermann ice tongue at ~2,200 cal yrs BP and its rapid expansion at ~600 cal yrs BP are preceded by a transition towards a southern ice arch regime with landfast ice formation in Nares Strait, suggesting a stabilizing effect of landfast sea ice on Petermann Glacier.



## 1 Introduction

Nares Strait, connecting the Lincoln Sea to the northern Baffin Bay, is an important conduit for sea ice, freshwater and heat
between the Arctic Ocean and the western North Atlantic. The annual flux of freshwater through Nares Strait, in liquid and
solid form, heavily depends on the seasonal formation of ice arches (Kwok et al., 2010; Münchow, 2016; Rasmussen et al.,
2010). Ice arches form when drift ice converges in a narrow passage between two landmasses. In Nares Strait their formation
depends primarily on the sea-ice thickness, local wind stresses, and atmospheric temperatures (Barber et al., 2001; Kwok et
al., 2010; Samelson et al., 2006). Ice arching inhibits sea-ice export from the Arctic Ocean and allows the formation of landfast
ice in Nares Strait, consisting of a mixture of multi-year drift ice, originating from the Arctic Ocean, and locally formed first-
year ice (Kwok, 2005; Kwok et al., 2010). Historically the formation of a northern and southern arch have been observed in
Robeson Channel and Smith Sound, respectively (Fig. 1) (Vincent, 2019). In recent decades, however, changes in the ice arch
configuration suggest a transition in Nares Strait sea-ice dynamics. Between 1979 and 2019, Nares Strait was blocked for sea-
ice passage on average 161 days per season with a consistent decrease of 2.1 days/year throughout this period (Vincent, 2019).
This is also associated with an emerging prominence of the northern arch (Vincent, 2019). In the winter of 2006/2007 both ice
arches failed to form for the first time in recorded history, causing sea ice to remain mobile in Nares Strait year-round (Kwok
et al., 2010; Vincent, 2019) (Fig. 2, Supplementary Fig. 1). The observed changes in Nares Strait sea-ice dynamics likely have
significant consequences for the export of multi-year sea ice from the Lincoln Sea (Vincent, 2019) and long-term Arctic sea-
ice loss. Additionally, the formation of the southern ice arch in Smith Sound is crucial for the annual opening of the North
Water Polynya (NOW) (Fig. 1) and the formation of landfast sea ice in Nares Strait (Barber et al., 2001). The latter has
important implications for the hydrographic structure in Nares Strait and its adjacent fjords in response to changing wind
stresses on surface waters in Nares Strait (Shroyer et al., 2015, 2017).

The water column in Nares Strait is characterized by cold and fresh Polar Water (PW) in the upper 50-100 m with warmer and
more saline modified Atlantic Water (AW) below, separated by a strong halocline (Johnson et al., 2011; Münchow et al.,
2014). Under landfast sea ice, Ekman transport causes eastward displacement of cold and fresh PW towards the Greenland
coast (Rabe et al., 2012; Shroyer et al., 2015, 2017). Conversely, mobile sea ice leads to westward Ekman transport of PW and
upwelling of AW in the east, increasing the oceanic heat flux to fjord systems along the Greenland coast of Nares Strait
(Münchow et al., 2007; Shroyer et al., 2017). Outlet glaciers draining into fjords in the north and northeast of Greenland,
commonly terminate in a floating ice tongue of variable length, with three glaciers terminating in an ice tongue >10 km (Hill
et al., 2017). One such glacier is Petermann Glacier (PG), draining about 4 % of the Greenland Ice Sheet (GrIS) (based on area
of the drainage basin (Rignot and Kanagaratnam, 2006)) into Hall Basin in Nares Strait (Fig. 1). Large calving events of PGs
floating ice tongue in 2010 and 2012 (Johannessen et al., 2013; Rückamp et al., 2019) were associated with a 10 % acceleration
of the glacier (Rückamp et al., 2019). Although the calving events have attracted considerable attention, submarine melting of





the ice tongue accounts for 80 % of the mass loss at PG, making it particularly sensitive to ice-ocean interactions (Cai et al.,

2017; Münchow et al., 2014; Rignot and Steffen, 2008; Rückamp et al., 2019).

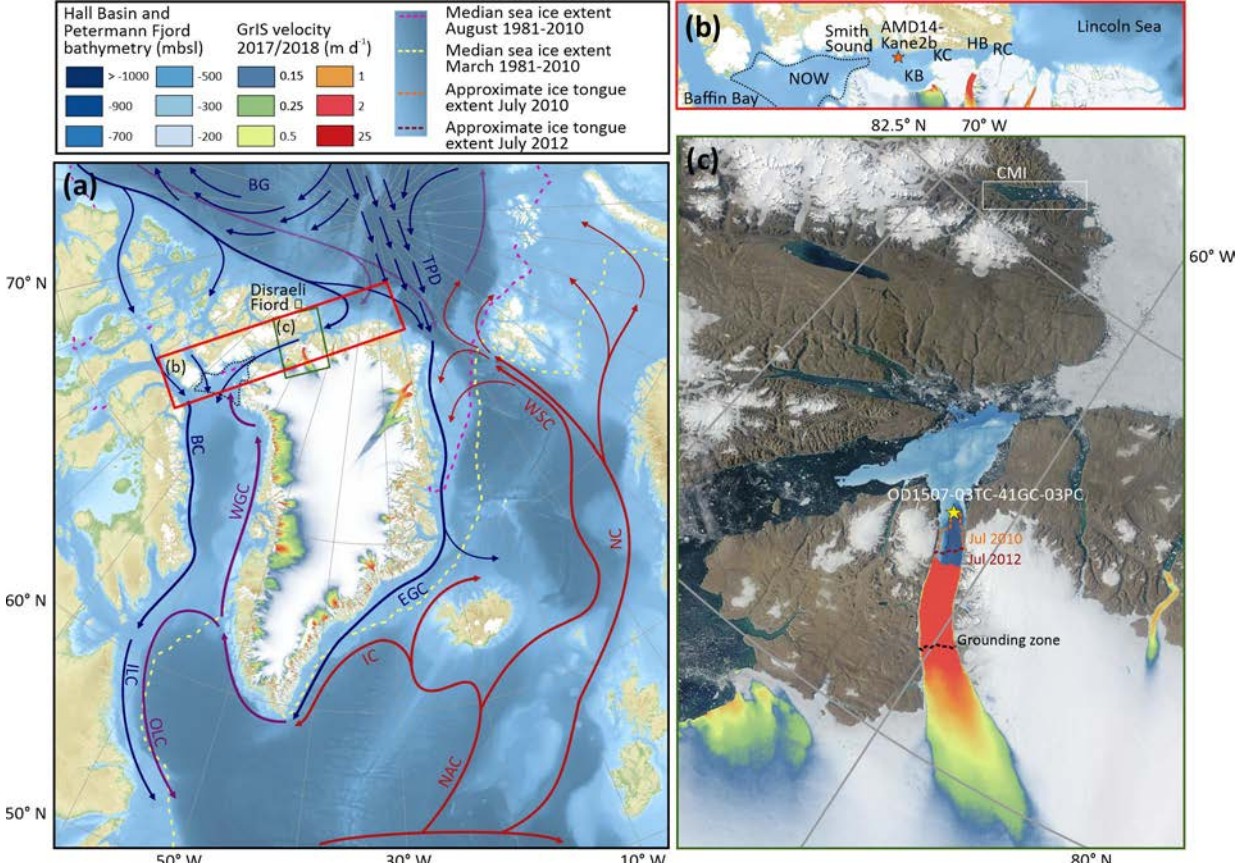

**Figure 1. Map of the study area. (a) Map of the North Atlantic, including major surface currents (BC: Baffin Current, BG: Beaufort Gyre, EGC: East Greenland Current, IC: Irminger Current, ILC: Inner Labrador Current, NAC: North Atlantic Current, NC: Norwegian Current, OLC: Outer Labrador Current, TPD: Trans Polar Drift, WGC: West Greenland Current, WSC: West**
**Spitsbergen Current), median August (dashed, pink) and March (dashed, yellow) sea ice extent from 1981-2010, the velocity of the Greenland Ice Sheet (GrIS) 2016/2017 (Nagler et al., 2015), and the location of Disraeli Fiord (Ellesmere Island). The approximate extent of the NOW is indicated with a black dashed line. (b) Close up of the Nares Strait connecting the Lincoln Sea to the northern Baffin Bay, consisting of the Robeson Channel (RC), Hall Basin (HB), Kennedy Channel (KC), Kane Basin (KB), and Smith Sound. Sediment core AMD14-Kane2b (red star) is located in the Kane Basin. The approximate extent of the NOW is indicated with a black**
**dashed line. (c) Close up of the Petermann Fjord and Petermann glacier (satellite image from 'NASA Worldview' (https://worldview.earthdata.nasa.gov/)) including the location of Clements Markham Inlet (CMI), the Hall Basin and Petermann Fjord bathymetry (Jakobsson et al., 2018), the velocity of the Greenland Ice Sheet (GrIS) 2016/2017 (Nagler et al., 2015), the grounding zone location of Petermann glacier (black, dashed), and the ice tongue extent in July 2010 (orange, dashed) and 2012 (red, dashed). The core location of OD1507-03TC-41GC-03PC is indicated with a yellow star.**

Submarine melt rates depend on the turbulent heat flux reaching the ice tongue/ocean interface. This is a function of the oceanic

heat content, determined by the inflow of modified AW to Petermann Fjord, and turbulent mixing underneath the ice tongue,

promoted by subglacial meltwater discharge (Cai et al., 2017; Washam et al., 2018). Thus, changes in Nares Strait sea-ice

dynamics, that modify AW inflow to Petermann Fjord, may affect the ice tongue stability on long timescales. Additionally,



earlier work has identified landfast sea ice in fjords around Greenland as an important mechanism stabilizing the calving front
of marine-terminating outlet glaciers, with the loss of landfast ice prolonging the calving season (Amundson et al., 2010; Carr et al., 2015; Robel, 2017; Todd and Christoffersen, 2014). Hence, in light of the recent doubling of mass loss from the GrIS (Shepherd et al., 2012) a detailed understanding of ocean-sea ice-glacier interactions on longer timescales is essential to improve projections for the contribution of the GrIS to future sea level rise (Fürst et al., 2015; Stocker et al., 2013).

In 2015, the Swedish Icebreaker *Oden* set out for *The Petermann 2015 Expedition* to improve our understanding of the
processes involved in ocean-sea ice-glacier interactions and the sensitivity of PGs floating ice tongue to Holocene climate change. A transect of sediment cores, extending from Hall Basin to underneath the Petermann ice tongue, was recovered (Reilly et al., 2019). Spatial differences in sediment facies associated with the presence/absence of the ice tongue allowed the reconstruction of the extent of the Petermann ice tongue over the last ~7,000 cal yrs BP (Reilly et al., 2019). Reilly et al. (2019) demonstrate that after the deglacial break-up of the ice tongue at ~6,900 cal yrs BP, no stable ice tongue existed in Petermann
Fjord for the largest part of the mid-Holocene, with a small ice tongue re-emerging at ~2,200 cal yrs BP, which advanced to its modern limits around 600 years ago.

Here, we focus on the Holocene evolution of sea-ice conditions in Petermann Fjord. The spliced sediment core OD1507-03TC-41GC-03PC (81.192° N; -62.023° E; 976 m water depth) located in outer Petermann Fjord (Fig. 1), offers a unique opportunity to study Nares Strait and local sea-ice dynamics, and how they influence the stability of PG. Sea-ice reconstructions are based
on source-specific Arctic sea-ice biomarkers (Belt, 2018). Measurements of total organic carbon (TOC), the carbon isotopic composition of TOC, sterol biomarkers and the benthic and planktonic foraminiferal abundance provide information with regard to marine primary productivity and terrestrial organic carbon input to Petermann Fjord across the Holocene. In combination with existing studies (England et al., 2008; Funder et al., 2011; Georgiadis et al., 2020) our results offer insights into the Holocene development of ice arches in Nares Strait. Importantly, this study demonstrates that the development of
more severe sea ice conditions in Petermann Fjord preceded major advances of the ice tongue, indicating a stabilizing effect on PG.

## 2. Regional oceanography

Nares Strait is an important conduit for heat and freshwater between the Arctic Ocean and the North Atlantic via the Baffin Bay. At its northern end, Robeson channel connects Hall Basin to the Lincoln Sea (Fig. 1) with water mass exchange controlled
by a 290 m deep sill (Münchow et al., 2011a; Washam et al., 2018). At the surface, Nares Strait is characterized by relatively fresh and nutrient-rich Polar Water (PW), in the upper 50 m (off Greenland) to 100 m (off Ellesmere Island) (Jones and Eert, 2004; Münchow et al., 2007). Geochemical tracers suggest that these waters are primarily of North Pacific origin, entering the Arctic Ocean via the Bering Strait, modified by river runoff and sea-ice melt (Jones and Eert, 2004; Münchow et al., 2007). Below, the water column is characterized by modified AW that has circulated through the Arctic Ocean (Jones and Eert, 2004).
Hydrographic surveys have shown that the modified AW in Nares Strait has warmed by $0.023 \pm 0.015$ °C per year between



2003 and 2009 (Münchow et al., 2011b). At its shallowest, in Kane Basin, Nares Strait is 220 m deep. This sill impedes the throughflow of the densest AW, suggesting that AW at the southern end of Nares Strait is derived from the north flowing West Greenland Current (Fig. 1) instead (Melling et al., 2001). The circulation in Nares Strait is dominated by a southward surface jet controlled by winds and along channel pressure gradients between the Lincoln Sea and northern Baffin Bay (Rabe et al.,

2010, 2012). The hydrographic structure in Nares Strait varies according to the predominant sea-ice state (Rabe et al., 2012; Shroyer et al., 2015, 2017). Modelling studies show that this is a response to surface stresses (Shroyer et al., 2015, 2017). Landfast sea ice exerts a northward drag at the ocean surface, resulting in eastward Ekman transport of cool and fresh PW and a westward shift of the main surface jet towards Ellesmere Island (Rabe et al., 2012; Shroyer et al., 2015, 2017). During the mobile sea-ice season southward wind stress and associated westward Ekman surface transport cause a displacement of cool

and fresh surface waters towards Ellesmere Island and upwelling of relatively warm and salty waters along the Greenland coast, while the main southward flow is concentrated in the center of the channel (Münchow et al., 2007; Shroyer et al., 2017). Nares Strait is covered by sea ice for around 10-11 months per year, with a 95% ice cover during winter months (Rasmussen et al., 2010). Summer break-up occurs in June/July, with renewed freeze up in late September/October. The flow of sea ice through Nares Strait is highest during fall and early winter (Kwok et al., 2010) with large interannual variability depending on

the formation and duration of the northern and southern ice arch. While the formation of either arch will block the export of Arctic sea ice through Nares Strait, only the formation of the southern arch leads to the opening of the NOW (Barber et al., 2001) and a complete freeze up of Nares Strait (Kwok, 2005; Kwok et al., 2010).

Petermann Fjord, connected to Nares Strait (Hall Basin) via a 350-450 m deep sill, is up to 1100 m deep and ~20 km wide (Jakobsson et al., 2018). It hosts a vast floating ice tongue, approximately 48 km long (from the grounding zone) with an

average width of 16.6 km and a thickness of 600 m at the grounding zone to 200 m at the terminus (Heuzé et al., 2017; Johannessen et al., 2013). The ice tongue flows at a speed of $1250 \pm 90$ m yr$^{-1}$ over the grounding zone, resulting in a calculated net glacial freshwater flux of 0.26 mSv (Heuzé et al., 2017). The formation of landfast ice in Petermann Fjord is somewhat independent of the formation of landfast ice in Nares Strait. Landfast ice in Petermann Fjord will also form when sea ice in Nares Strait remains mobile throughout the winter, although the sea-ice state in Nares Strait likely influences the timing of

sea-ice break-up in Petermann Fjord in spring/summer (Fig. 2, Supplementary Fig. 1). Additionally, the sea-ice dynamics in Nares Strait have important implications for the primary productivity regime in Petermann Fjord, with ice edge conditions in spring/summer during years with mobile sea ice in Nares Strait (Supplementary Fig. 1). The hydrographic structure in Petermann Fjord is closely linked to that in Nares Strait with colder, fresher PW overlaying modified AW (Johnson et al., 2011; Münchow et al., 2014). Bottom waters in Petermann Fjord are renewed by episodic spillover of AW from Hall Basin,

with bottom water properties in Petermann Fjord resembling those at ~380 m water depth in Hall Basin (effective sill depth) (Johnson et al., 2011). In general, the circulation in Petermann Fjord resembles an estuarine model, with outflow of buoyant, meltwater-enriched surface waters along the northeast side of the fjord and inflow of modified AW below, concentrated along the southwestern side of the fjord mouth (Heuzé et al., 2017; Johnson et al., 2011; Washam et al., 2018). In the fjord mouth, eddy structures can enhance the exchange between Hall Basin and Petermann Fjord. Eddies are stronger and more stable during



summer, when sea ice in Nares Strait is mobile (Johnson et al., 2011; Shroyer et al., 2017). Modelling studies indicate that the displacement of water masses in response to the prevailing sea-ice state in Nares Strait penetrates into Petermann Fjord, with enhanced inflow of warmer, saltier AW during times of mobile sea-ice (Shroyer et al., 2017). Additionally, a stronger circulation in the fjord, driven by enhanced subglacial runoff during summer months, increases the transport of AW to the ice tongue cavity and the turbulent mixing of AW toward the base of the ice tongue (Cai et al., 2017; Washam et al., 2018). In response to warming of AW in Nares Strait (Münchow et al., 2011b), a 0.2°C warming of AW in Petermann Fjord has been observed from 2002 to 2016 (Washam et al., 2018). In combination, the greater oceanic heat flux and strengthened under ice currents cause enhanced submarine melting and non-steady state thinning of the ice tongue (Cai et al., 2017; Washam et al., 2018, 2019).

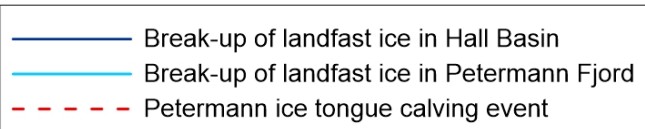

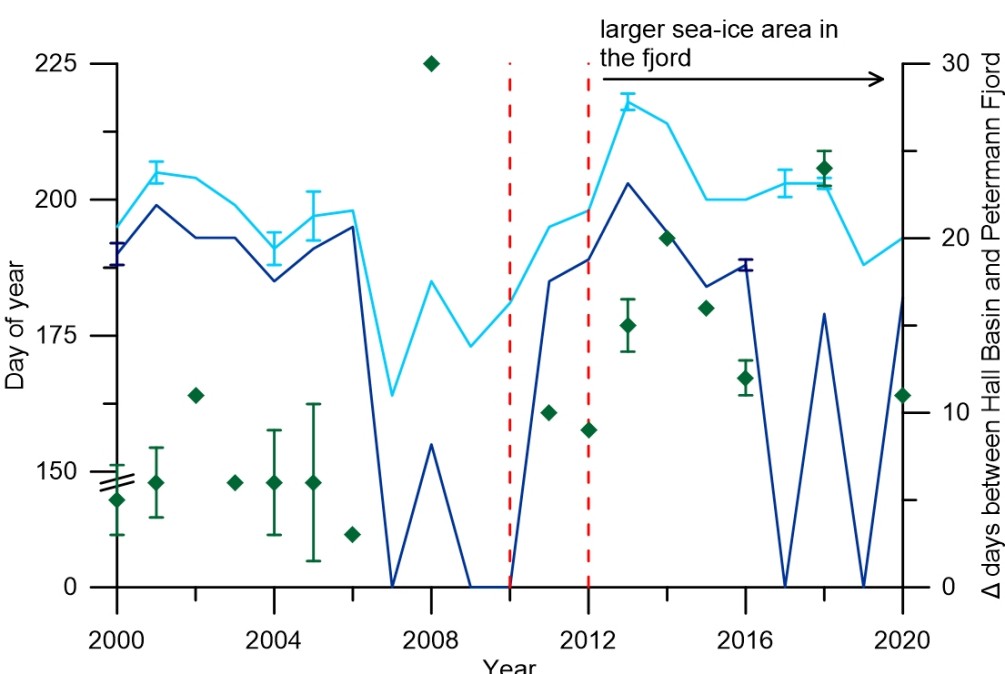

**Figure 2. Time series of approximate landfast sea-ice break-up in Petermann Fjord (light blue) and Hall Basin (dark blue) from 2000-2020 estimated from https://worldview.earthdata.nasa.gov/. Where the Hall Basin record has a day of year of zero, landfast sea ice did not form throughout the entire winter. Where cloud cover inhibited the exact determination of the day of landfast sea-ice break-up, the average between the last clear day with landfast ice and the first clear day following ice break-up was used (Hall Basin: 2000,2016; Petermann Fjord: 2001, 2004, 2005, 2013, 2017, 2018). Error bars indicate the timespan where cloud cover inhibited clear determination of the sea-ice state. The dashed vertical lines indicate years of substantial ice tongue calving in Petermann Fjord. The green diamonds indicate the difference between ice break-up in Hal Basin and Petermann Fjord in days. After 2010/2012 the differences increases due to the larger sea-ice area in Petermann Fjord following the retreat of the ice tongue.**





## 3. Materials and methods

### 3.1 Sediment core OD1507-03TC-41GC-03PC

OD1507-03TC-41GC-03PC represents a spliced record of a trigger core (TC), a gravity core (GC), and a piston core (PC) recovered in outer Petermann Fjord as part of *The Petermann 2015 Expedition* (Fig. 1) (Table 1). The splice has a total length of 556 cm and was recovered 80 km from the 2015 PG grounding zone at an average water depth of 976 m (Reilly et al., 2019). Based on Computed Tomography (CT) scans, continuous sections with minimal disturbance were chosen for the splice and correlation was performed using X-Ray Fluorescence (XRF) Ti/Ca ratios, CT slice images, and CT numbers (Reilly et al.,

2019) (Supplementary Table 1). Three sedimentary units are distinguished in OD1506-03TC-41GC-03PC, based on physical properties and XRF elemental data (Reilly et al., 2019). Sedimentary unit 3 (ca. 518-555 cm) is a massive diamict composed of a sandy mud and abundant coarse clasts and XRF Ti/Ca ratios around 0.05 (Reilly et al., 2019). Unit 2, a clayey mud with faint laminations and no or low concentrations of coarser material, is found between ca. 398-518 cm (Reilly et al., 2019). The topmost lithofacies, unit 1 (0-398 cm), is a bioturbated clayey mud with isolated sand and coarser particles (Reilly et al., 2019).

Based on the XRF Ti/Ca ratios and the abundance of coarse material, unit 1 has been divided into three subsections (A (0-53 cm), B (53-164 cm), C (164-398 cm)) (Reilly et al., 2019).

**Table 1. Parameters of sediment cores used in the OD1507-03TC-41GC-03PC splice (Reilly et al. 2019).**

| Expedition | Deployment # | Core type | Latitude (°) | Longitude (°) | Water depth (m) | Length (cm) |
|---|---|---|---|---|---|---|
| OD1507 | 03 | PC/TC | 81.190 | -62.068 | 960 | 606/95.3 |
| OD1507 | 41 | GC | 81.194 | -61.977 | 991 | 440 |

The age model for OD1507-03TC-41GC-03PC, established by Reilly et al. (2019), is based on radiocarbon ($^{14}$C) dating of benthic and planktonic foraminifers and calibration of radiocarbon ages using the Marine13 curve (Reimer et al., 2013) and MatCal MATLAB tools (Lougheed and Obrochta, 2016). The ΔR value was constrained using Paleosecular Variation (PSV) stratigraphy, with the best fit determined for a constant ΔR choice of 770 years (Reilly et al., 2019). No ages were determined for the interval between 408 cm and 556 cm, due to lack of radiocarbon dates. Regional constraints, however, suggest that

these sediments are younger than 7,600 years, corresponding to the timing of the retreat of PG into the fjord (Jakobsson et al., 2018).

### 3.2 Sea-ice biomarker methodology

Reconstructions of past sea-ice conditions in Petermann Fjord rely on the identification of source-specific Arctic sea-ice biomarkers, together with the identification of common sterol biomarkers, TOC, and planktonic and benthic foraminiferal

abundances. This is a qualitative method to determine past sea-ice dynamics. For clarity, past sea-ice conditions are classified

 

into four categories: ice free, reduced seasonal sea ice, enhanced seasonal sea ice, and near-perennial (in order of increasing average sea-ice concentration for a given area).

Source-specific sea-ice biomarkers include a mono- and a di-unsaturated highly branched isoprenoid (HBI), termed IP$_{25}$ (ice proxy with 25 carbon atoms) and HBI II (Belt, 2018; Belt et al., 2007). IP$_{25}$ is produced by a number of spring sea-ice dwelling

diatoms, including *Haslea spicula*, *H. kjellmanii*, and *Pleurosigma stuxbergii* var. *rhomboides* (Brown et al., 2014). Thus, its presence in Arctic marine sediments provides evidence for past seasonal sea-ice occurrence, while the absence of IP$_{25}$ occurs in year-round ice-free environments as well as under perennial sea-ice cover (Belt, 2018; Belt et al., 2007; Brown et al., 2014; Navarro-Rodriguez et al., 2013; Xiao et al., 2015). Salinity changes might exert an additional control on IP$_{25}$ production, with Ribeiro et al. (2017), showing that its production appears to be suppressed by meltwater in fjords of North Eastern Greenland.

Given its co-production (Brown et al., 2014), HBI II co-varies with IP$_{25}$ in the Arctic realm. Thus, the typically higher HBI II concentrations can provide additional information during times of low/absent sedimentary IP$_{25}$.

**Table 2. Overview of the dominant sources of different sterol biomarkers measured in this study.**

| Biomarker | Predominant source organisms | References |
|---|---|---|
| Cholesterol | - In the marine realm: zoo- and phytoplankton<br>- Also occurs in terrestrially derived organic carbon | Meyers and Ishiwatari (1993), Volkman (1986), Volkman (2003), Volkman et al. (2000), Yunker et al. (1995) |
| Brassicasterol | - Often the most abundant sterol in marine diatoms<br>- In lower relative concentrations found in freshwater algae | Volkman (1986), Volkman (2003), Volkman et al. (2000), Yunker et al. (1995) |
| Dinosterol | - Major sterol in most dinoflagellate species<br>- Also found in other marine microalgae (e.g. diatoms) | Belicka et al. (2004), Volkman (2003), Volkman et al. (2000) |
| Campesterol | - Dominant sterol in vascular plants<br>- In minor amounts synthesized by marine microorganisms | Rontani et al. (2014), Volkman (1986), Yunker et al. (1995) |
| β-sitosterol | - Dominant sterol in vascular plants<br>- In minor amounts synthesized by marine microorganisms | Rontani et al. (2014), Volkman (1986), Yunker et al. (1995) |

In recent years, biomarker-based reconstructions of past sea-ice dynamics have commonly also reported a tri-unsaturated HBI (HBI III), produced by diatoms characteristic of the spring sea-ice edge bloom in the marginal ice zone (MIZ) (Belt et al., 2015, 2017; Smik et al., 2016). In Petermann Fjord, however, the identification of the HBI III peak in chromatograms from gas chromatography-mass spectrometry (GC-MS) was compromised due to interference with another peak (Supplementary Fig. 2). This is specific to sediment samples from Petermann Fjord, as HBI III could be identified in reference sediment from

Young Sound extracted alongside the samples from Petermann Fjord (see 3.3.2). Instead, we use a range of sterol biomarkers





(campesterol, cholesterol, brassicasterol, dinosterol, and β-sitosterol) and the abundance of benthic and planktonic foraminifers to gain insights into the regional primary productivity regime and the addition of terrestrial organic carbon to Petermann Fjord. Sterols are common compounds in eukaryotic cell membranes, abundant in both marine and terrestrial organic carbon. This complicates their use as unambiguous tracers of a specific organism and/or environmental regime (Belt and Müller, 2013;

Rontani et al., 2014; Volkman, 1986). Nonetheless, the relative abundance of certain sterols can vary according to the predominant organic carbon source (Rontani et al., 2014; Volkman, 1986; Yunker et al., 1995) (Table 2).

Brassicasterol and dinosterol are commonly reported in studies of past sea-ice variability and have been found in regions with differing sea-ice conditions (Xiao et al., 2015). Depending on the predominant primary producers, high concentrations of brassicasterol and dinosterol are reported from ice-free regions, the MIZ, and under varying concentrations of seasonal sea ice

(Xiao et al., 2015). Sterol concentrations are low under perennial sea ice in the Arctic Ocean, where all primary productivity is impeded due to limited light availability (Xiao et al., 2015).

Here we use brassicasterol, dinosterol, and cholesterol to represent phytoplankton primary productivity in Petermann Fjord, with different relative concentrations of these sterols, potentially indicating changes in the ecosystem composition. Campesterol and β-sitosterol, on the other hand, are used to gain insight into enhanced terrestrial organic carbon input. The

multiproxy study of $IP_{25}$, HBI II, sterol biomarkers, and the benthic and planktonic foraminiferal abundance (see section 3.4), thus allows interpretations of past sea-ice dynamics and terrestrial versus marine organic carbon input to Petermann Fjord.

### 3.3 Analysis of total organic carbon (TOC), carbon isotopic composition of TOC ($\delta^{13}C_{org}$), and sea-ice related biomarkers in sediments from OD1507-03TC-41GC-03PC

OD1507-03TC-41GC-03PC is stored at the Oregon State University Marine and Geology Repository at 2.8°C. The core was

sampled approximately every 5 cm for the analysis of different organic constituents. Samples were immediately frozen, shipped, and freeze-dried (-45 °C; 0.2 mbar; 48 h) at Aarhus University using a Christ Alpha 1-4 LSC freeze drier. Subsequently the samples were homogenized with a dichloromethane (DCM) cleaned pestle and mortar.

### 3.3.1 Total organic carbon (TOC) quantification and analysis of carbon isotopic composition of TOC ($\delta^{13}C_{org}$)

For TOC and $\delta^{13}C_{org}$ 10 mg of homogenized sample material was weighed into an Ag capsule (Elemental Microanalysis).

Depending on available sample material, duplicates or triplicates were prepared to test for reproducibility. Additionally, a soil standard (10 mg; Elemental Microanalysis Soil Standard (Sandy) OAS 133506; 0.76 % $w/w$ TOC) and blanks were added every ~15 samples. To remove the inorganic carbon, 35% HCl was added one drop at a time until no further reaction was observed (approximately 4 drops per sample, standard, and blank). The samples were then dried on a hotplate (50°C) overnight. The Ag capsules were folded and placed inside a Sn capsule (Elemental Microanalysis), which was packed tightly and stored

in a desiccator until analysis.

The stable isotope $\delta^{13}C_{org}$ and % wt TOC were measured using a continuous-flow IsoPrime IRMS coupled to an Elementar PyroCube elemental analyser at the Aarhus AMS Centre (AARAMS), Aarhus University, Denmark. $\delta^{13}C_{org}$ is reported in ‰





versus Vienna Pee Dee Belemnite (VPDB). An in-house standard Gel-A was used as primary standard yielding ±0.2‰ and
±0.3‰ for carbon and nitrogen analysis respectively. Further, secondary in-house and international standards were used to
check the normalization to the VPDB. The mean reproducibility using 10 duplicate samples is ±0.03% and ±0.4‰ for % wt
TOC and $\delta^{13}C_{org}$ respectively. For both acid pre-treated and non-acid pre-treated Elemental Microanalysis Soil Standard
(Sandy) OAS 133506 samples the reproducibility of 7 samples is ±0.02% wt TOC. The $\delta^{13}C_{org}$ reproducibility non-acid
pretreated Elemental Microanalysis Soil Standard (Sandy) OAS 133506 samples is ±0.1‰ whereas acid pre-treated samples
show a mean reproducibility of ±0.5‰.


TOC concentrations are reported in weight % (wt.%) and TOC fluxes (µg cm$^{-2}$ yr$^{-1}$) are derived using the samples individual
dry bulk density and linear sedimentation rates (LSR), which were calculated based on the datums from $^{14}$C analysis of benthic
and planktonic foraminifera (Reilly et al., 2019). Dry bulk densities were calculated using the samples respective volume and
dry weight.

**3.3.2 Lipid biomarker extraction and analysis**

Biomarkers were extracted from 6.00±0.03 g freeze-dried sediment. A procedural blank and a reference sediment sample (~3
g) with known biomarker concentrations were added to each extraction batch (n = 12). 9-octylheptadec-8-ene (9-OHD) and
5α-androst-16-en-3α-ol (0.1 µg) were added to each sample, reference sediment, and blank, as internal standards for HBI and
sterol quantification, respectively. The samples were extracted using saponification (5% potassium hydroxide (KOH) in
methanol (MeOH):H$_2$O (9:1, $v/v$); 70°C, 1 h) followed by extraction of the non-saponifiable lipids into hexane (3 x 2 mL).
According to polarity, the different lipid classes were separated using silica column chromatography. Nonpolar lipids, such as
IP$_{25}$ and HBI II were eluted with hexane, while the more polar sterols were eluted with DCM:MeOH (1:1, $v/v$) (~7 mL, each).
HBI fractions were further purified using silver nitrate silica column chromatography (AgNO$_3$ on SiO$_2$, ~10 wt.% of labeling),
where the saturated hydrocarbons were eluted with hexane (2 mL) and the unsaturated compounds, including the HBIs, were
eluted with acetone (7 mL), dried (N$_2$; 25°C) and transferred to GC-MS vials fitted with 300 µL inserts. The sterol fractions
were derivatised using N,O- Bis(trimethylsilyl)trifluoroacetamide (50 µL, 70 °C, 1 h) and transferred to 1.5 mL GC-MS vials.
All biomarker samples were analyzed at Aarhus University using an Agilent 7890B GC fitted with an HP-5ms Ultra Inert
column (30 m x 250 µm x 0.25 µm) coupled to a 5977A series mass selective detector and equipped with a Gerstel multipurpose
sampler (MPS). Prior to analysis HBI and sterol extracts were diluted with 50 µL hexane and 0.500 mL DCM, respectively,
using the MPS system. For GC-MS operating conditions see Supplementary Table 2. Following Belt (2018) the identification
of individual lipids is based on their characteristic retention indices and mass spectra. Quantification is achieved by comparing
the integrated peak area (PA) of the selected ion for each biomarker (Supplementary Table 2) to the PA of the respective
internal standard (Belt et al., 2012) under consideration of an instrumental response factor (based on the reference sediment)
and the mass or the TOC concentrations of the sediment extracted (Belt et al., 2012). Due to unknown concentrations of
dinosterol in the reference sediment, an individual response factor could not be determined for this compound. Instead, the





average response factor for brassicasterol and cholesterol was used. Thus, while the relative dinosterol concentrations and trends hold true, absolute concentrations might not be accurate. In addition to the biomarker concentrations in ng g$^{-1}$ of dry sediment (ng g$^{-1}$ sed) and µg g$^{-1}$ TOC, biomarker fluxes were calculated using the samples individual dry bulk density and linear sedimentation rates (LSR). Dry bulk densities were calculated using the samples respective volume and dry weight.

Fluxes are reported in ng cm$^{-2}$ yr$^{-1}$ and are interpreted alongside biomarker concentrations to avoid bias related to large jumps in the LSR.

### 3.3 Planktonic and benthic foraminiferal abundances

Benthic and planktonic foraminifers respond to a multitude of environmental factors that determine their abundance. Additionally, postmortem dissolution of carbonate shells can affect the number of shells preserved in the sediments. In the

high Arctic, the dominant environmental factors include the water mass properties, the presence of glaciers and sea ice and their respective effects on marine primary productivity and food availability (Jennings et al., 2020). Sea ice affects the biogeographical distribution of benthic and planktonic foraminifers via its influence on the ocean circulation and water mass properties, and its effect on the amount and seasonality of the organic carbon flux (Jennings et al., 2020). Under perennial sea-ice cover, primary productivity is greatly reduced as a result of the restricted light conditions in the surface ocean. Conversely,

the spring sea-ice edge is a region of enhanced productivity, related to the stratifying effect of the melting sea ice together with the release of nutrients into the water column. Thus, while the abundance of planktonic foraminifera directly responds to the amount of sea ice cover, the benthic foraminiferal abundance is indirectly influenced by sea ice via its effects on the water column and organic carbon flux. In this study, planktonic foraminiferal abundance is used as an indicator of predominantly open water primary productivity, while benthic foraminiferal abundance is a more general indicator of organic carbon flux to

the seafloor.

The benthic and planktonic foraminiferal abundance in OD1507-03TC-41GC-03PC was determined on 58 samples with an average resolution of 10 cm. The benthic foraminiferal counts include both calcareous and agglutinated species. Where sufficient core material was available the sample depths correspond to the TOC and biomarker samples. The foraminiferal samples were weighed and wet sieved at 63 µm. The >63 µm was counted wet, submerged in a 'storage' solution of 70 %

distilled water and 30 % ethanol with baking soda to preserve fragile calcareous and agglutinated tests. A wet splitter was used when necessary to achieve a count of at least 200-300 benthic foraminifers. Planktonic foraminifers were counted in the benthic split. Equivalent dry weights of the foraminiferal samples were calculated using the wet weights of the foraminiferal samples and the wet and dry weights of other samples from the same depths. That way the numbers of benthic and planktonic foraminifers per gram of dry sediment could be calculated without drying the samples. Foraminiferal fluxes were calculated

using the samples dry bulk densities and LSR and reported in specimens cm$^{-2}$ yr$^{-1}$, were the depth of the biomarker and foraminiferal samples were not the same, DBD for foraminifera samples was determined by linearly interpolating between adjacent samples.



## 4. Results

### 4.1 Benthic and planktonic foraminiferal concentrations

The benthic and planktonic foraminiferal abundances vary between 0.2 and 429.3 specimens per g sediment and 0 and 92.0 specimens per g sediment, respectively (Fig. 3D, E). This corresponds to benthic and planktonic fluxes of 2.5-32.3 specimens per cm$^2$ per year and 0.1-8.1 specimens per cm$^2$ per year (Fig. 3D, E). In sedimentary unit 2 and 3, benthic and planktonic foraminiferal abundances are low. Both benthic and planktonic abundances increase near the unit 2/unit 1C boundary, at 404 cm and 384 cm depth, respectively. Unit 1C is characterized by the highest overall foraminiferal abundances and fluxes with

planktonic abundance and fluxes peaking between 304 cm and 312 cm prior to the benthic abundance, which peaks at 254 cm (Fig. D, E3). Planktonic foraminifera abundances and fluxes decline from 250 cm in unit 1C and continue at low values through unit 1B and are nearly absent in unit 1A (Fig. 3E). Benthic foraminiferal abundances and fluxes are more variable, but reach their lowest abundance in unit 1A (Fig. 3 D).

### 4.2 Total organic carbon (TOC) concentration and TOC carbon isotopes (δ$^{13}$C$_{org}$)

The overall amount of total organic carbon (TOC) in OD1507-03TC-41GC-03PC is very low, varying between 0.1 wt.% and 0.3 wt.% (Fig. 3C). Between 289-394 cm and at 109 cm, several samples with high TOC (0.3-1.8 wt.%) and low carbon isotopic composition of the organic carbon (δ$^{13}$C$_{org}$; -26.2 ‰ to -30.4 ‰), suggest incomplete inorganic carbonate removal during sample processing. The interval between 289 cm and 394 cm is associated with high Ti/Ca ratios and IRD delivery to the core site, indicating increased glacial sedimentation in outer Petermann Fjord (Reilly et al., 2019) (Fig. 3A), which might

have been associated with input of detrital carbonates. Thus, these samples have been removed from the TOC and δ$^{13}$C$_{org}$ record (Fig. 3B and C).

The lowest TOC is encountered in sedimentary unit 3, characterized as grounding zone proximal sedimentation (Reilly et al., 2019), followed by a sharp increase prior to the boundary of sedimentary unit 3 and 2 at 520 cm depth. The lower unit 2 is characterized by large variability in the TOC content with two peaks at 520 cm and 490 cm, while the upper sediments of the

same unit have a more stable TOC content around 0.15 wt.% (Fig. 3C). From 409 cm a steady increase of TOC into sedimentary unit 1C is observed. The TOC content in sedimentary unit 1C, 1B, and 1A varies between 0.17 and 0.27 wt.% with local maxima around 250 cm and 94 cm and minima at 159 cm and 45 cm (Fig. 3C). TOC fluxes throughout these units vary between 109 and 422 µg cm$^{-2}$ yr$^{-1}$ with maximum fluxes from 94-194 cm, corresponding to the interval of maximum LSR. After 94 cm TOC fluxes decrease towards the boundary of sedimentary unit 1B and 1A with minimum fluxes recorded at 54 cm and flux

values similar to unit 1C throughout unit 1A (Fig. 3C).

The δ$^{13}$C$_{org}$, measured on the same samples as TOC, varies between -24.5 ‰ and -28.2 ‰. The lowest values are observed in sedimentary unit 3 and 2 with minima at 490 cm and 530 cm and a relatively steady rise throughout the latter (Fig. 3B). Sedimentary unit 1C, 1B, and 1A are marked by δ$^{13}$C$_{org}$ values between -26.4 ‰ and -24.5 ‰ with local minima at 249 cm, 159 cm, and 10 cm (Fig. 3B).



### 4.3 HBI biomarker concentrations


$IP_{25}$ and HBI II were analysed in 100 samples with an average depth resolution of 5.5±1.8 cm and temporal resolution of 99±45 years. Both biomarkers were present in all samples, with an overall range of 0.3-30.0 ng g$^{-1}$ sed for $IP_{25}$ and 0.6-83.0 ng g$^{-1}$ sed for HBI II and fluxes of 0.2-3.0 ng cm$^{-2}$ yr$^{-1}$ and 0.3-8.0 ng cm$^{-2}$ yr$^{-1}$, respectively (Fig. 3F, G). HBI concentrations normalized to the amount of sediment mirror the biomarker concentrations normalized to the TOC content of the samples

(Supplementary Fig. 3). In line with co-production of these two biomarkers (Brown et al., 2014) a significant positive correlation was found ($R^2 = 0.95$ [0.84; 0.97], n=100). Sedimentary unit 3, characterized as grounding zone proximal sediments by Reilly et al. (2019), is marked by very low $IP_{25}$ and HBI II concentrations (Fig. 3F, G). Both $IP_{25}$ and HBI II concentrations increase stepwise, with a first increase at 515 cm corresponding to the transition from sedimentary unit 3 to 2, characterized by laminated sediments with very little/no IRD input, signifying the presence of an extensive ice tongue in Petermann Fjord

(Reilly et al., 2019). The second increase in biomarker concentrations at 405 cm precedes the sedimentary unit boundary (1C/2) by ~5 cm and is followed by an interval of highly variable $IP_{25}$ and HBI II concentrations and fluxes until 320 cm (Fig. 3F, G). From 260 cm, a steep increase in $IP_{25}$ and HBI II concentrations and fluxes culminates in peak concentrations at 199 cm followed by peak fluxes at 189 cm (Fig. 3F, G). Subsequently a two-step decrease in all HBI concentrations and fluxes is observed between 154 cm and 199 cm, spanning the transition from sedimentary unit 1C to 1B at ~165 cm (Reilly et al., 2019)

(Fig. 3F, G). At 154 cm, a sharp increase in $IP_{25}$ marks the onset of a second interval with sustained high concentrations between 154 cm and 70 cm. $IP_{25}$ fluxes increase simultaneously, but decrease prior to the concentrations at 89 cm in connection with a decrease in the LSR (Fig. 3A, G). HBI II concentrations remain relatively low throughout lower sedimentary unit 1B, while the fluxes follow the $IP_{25}$-flux trend (Fig. 3F, G). At 70 cm, however, both $IP_{25}$ and HBI II concentrations and fluxes drop sharply, preceding the transition from sedimentary unit 1B to 1A by ~7 cm (Fig. 3F, G). HBI biomarker concentrations

and fluxes remain low throughout unit 1A, with a minor increase around 25 cm (Fig. 3F, G).

The ratio of HBI II and $IP_{25}$ ($DIP_{25}$) has previously been used as an indicator for sea surface temperature (SST) and thus as a tracer of warmer water masses (Hörner et al., 2016; Xiao et al., 2013), in line with higher temperatures being more favorable for the synthesis of double bonds. Other studies, however, did not find a relationship between $DIP_{25}$ and SST and propose instead that a steady $DIP_{25}$ reflects stable sea-ice conditions, while a variable $DIP_{25}$ indicates more unstable sea-ice conditions

(Belt and Müller, 2013; Cabedo-Sanz et al., 2013). In outer Petermann Fjord, the $DIP_{25}$ ratio is low throughout sedimentary unit 3 and rises before the unit3/unit 2 boundary (Supplementary Fig. 4). This is followed by the highest recorded $DIP_{25}$ values between 430 cm and 520 cm in unit 2. At 430 cm a sharp decline in $DIP_{25}$ values marks the onset of a 110 cm long section with relatively low but variable $DIP_{25}$ values ($\sigma^2 = 0.41$). At 320 cm, a small increase in $DIP_{25}$ is followed by a relatively steady decline throughout upper unit 1C, unit 1B, and unit 1A (Supplementary Fig. 4). This coincides with a reduced variance in the

data ($\sigma^2 = 0.26$).





**Figure 3. HBI biomarker, TOC, and foraminifers results at OD1507-03TC-41GC-03PC. From the top to the bottom: (a) Linear sedimentation rates (LSR; black), the XRF Ti/Ca ratio (grey), and the CT >2 mm clast index (brown) (Reilly et al., 2019). (b) TOC carbon isotopic values (δ$^{13}$C$_{org}$ ‰, VPDB; light green). (c) TOC fluxes (green, filled in area) and TOC concentrations (green line).**
**380** **(d) Benthic foraminiferal fluxes (purple, filled in area) and benthic foraminiferal abundance in individuals per g sediment (purple line with diamonds). (e) Planktonic foraminiferal fluxes (crimson, filled in area) and plankonic foraminiferal abundance in individuals per g sediment (crimson line with diamonds). (f) HBI II fluxes (turquois, filled in area) and absolute concentration (turquois line with dots) normalised to the amount of extracted sediment (ng g$^{-1}$ sed.). (g) IP$_{25}$ fluxes (blue, filled in area) and absolute concentration (blue line with dots) normalised to the amount of extracted sediment (ng g$^{-1}$ sed.). The vertical background fill indicates**
**385** **the lithological units (1A-1C, 2, 3) at OD1507-03TC-41GC-03PC (Reilly et al., 2019).**

**4.4 Sterol biomarker concentrations**

Sterol biomarkers were measured in 94 samples with a depth and temporal resolution of 5.7±2.3 cm and 102±51 years, respectively. We measured brassicasterol, dinosterol, and cholesterol (hereinafter grouped as marine sterols) and campesterol and β-sitosterol (hereinafter grouped as terrestrial sterols). All sterols were present consistently throughout the core (Table 3).

There are minor differences in the temporal evolution of sterol concentration normalized to the amount of sediment extracted and to the TOC content of the samples (Supplementary Fig. 3). These become especially apparent in the dinosterol concentrations between 400-500 cm, the campesterol concentrations between 400-500 cm and the β-sitosterol concentrations. The differences are focused on the lower part of the record, where Petermann Fjord is influenced by enhanced influx of terrestrial organic carbon (see section 5.1). Thus, we use normalization to extracted sediment mass and the resulting fluxes to

make inferences about past environmental changes. One sample at ~319 cm has β-sitosterol concentrations three times higher than the average, while all other sterol concentrations fall within the reported range (Supplementary Fig. 5). It is not clear why the β-sitosterol concentrations are so high in this specific sample. Thus, we have excluded this sample from all further analyses and interpretations. Cholesterol is significantly correlated with dinosterol ($R^2$ = 0.90 [0.64; 0.98], n = 93) and brassicasterol ($R^2$ = 0.93 [0.75; 0.98], n = 93), in line with a predominantly marine source of cholesterol in Petermann Fjord. Further, the marine

sterols show a good correlation with the IP$_{25}$ concentrations (IP$_{25}$-brassicasterol: $R^2$ = 0.82 [0.62; 0.92], n = 93, IP$_{25}$-cholesterol: $R^2$ = 0.89 [0.69; 0.96], n = 93, IP$_{25}$-dinosterol: $R^2$ = 0.82 [0.56; 0.93], n = 93), suggesting an important role of sea ice for the regional marine primary productivity throughout the record.

Similar to the HBIs, the concentrations of all sterols increase at the transition from sedimentary unit 3 to 2 with the most significant increase observed in β-sitosterol and campesterol (Fig. 4B, C). Both terrestrial sterols are characterized by a double

peak throughout sedimentary unit 2, followed by a decline at the 1C/2 boundary, while the concentrations of marine sterols remain relatively stable throughout this interval (Fig. 4). Throughout unit 1C an increase in the cholesterol and dinosterol concentrations is observed from 350 cm, while their fluxes, alongside all other sterol concentrations remain low until 260 cm (Fig. 4D, E). Peak concentrations are reached between 199 cm and 189 cm, followed by a slight decrease in all sterol concentrations and a recovery in their concentrations during lower sedimentary unit 1B (Fig. 4). Simultaneously, sterol fluxes

are at their highest between 94 cm and 194 cm, corresponding to the interval of maximum LSR (Fig. 4). While the sterol fluxes decrease from 94 cm, concentrations peak from 74-84 cm, followed by a sharp decrease prior to the transition from sedimentary unit 1B to 1C. Unit 1C is characterized by overall low sterol concentrations and fluxes (Fig. 4).





**Figure 4. Sterol biomarker results at OD1507-03TC-41GC-03PC. From the top to the bottom: (a) Marine sterol index (sum of marine sterols/sum of all sterols) (purple line). (b) β-sitosterol fluxes (brown, filled in area) and absolute concentration (brown line with dots) normalised to the amount of extracted sediment (ng g⁻¹ sed.). (c) Campesterol fluxes (light brown, filled in area) and absolute concentration (light brown line with dots) normalised to the amount of extracted sediment (ng g⁻¹ sed.). (d) Cholesterol fluxes (orange-brown filled in area) and absolute concentration (orange-brown line with dots) normalised to the amount of extracted sediment (ng g⁻¹ sed.). (e) Dinosterol fluxes (dark green, filled in area) and absolute concentration (dark green line with dots) normalised to the amount of extracted sediment (ng g⁻¹ sed.). (f) Brassicasterol fluxes (light green, filled in area) and absolute concentration (light green line with dots) normalised to the amount of extracted sediment (ng g⁻¹ sed.). The vertical background fill indicates the lithological units (1A-1C, 2, 3) at OD1507-03TC-41GC-03PC (Reilly et al., 2019).**

For further insight into the environmental factors driving sterol variability, the marine sterol index (sum of marine sterols/sum of all sterols) was determined (Stein et al., 2017). This indicates that sedimentary unit 3, 2 and the lowermost unit 1C are dominated by terrestrial sterols, with a slight increase in the relative concentration of marine sterols just prior to the 2/1C boundary (Fig. 4A). The relative concentration of marine sterols increases between 314 cm and 59 cm, followed by a decrease at the boundary of sedimentary unit 1A/1B (Fig. 4A).

**Table 3. Ranges of sterol biomarker concentrations and fluxes in OD1507-03TC-41GC-03PC**

| Biomarker | Min. concentration (ng g⁻¹ sed) | Max. concentration (ng g⁻¹ sed) | Min. flux (ng cm⁻² yr⁻¹) | Max. flux (ng cm⁻² yr⁻¹) |
|---|---|---|---|---|
| Brassicasterol | 17.6 | 771.1 | 7.1 | 91.0 |
| Dinosterol | 2.8 | 65.7 | 1.3 | 10.0 |
| Cholesterol | 62.4 | 1796.2 | 22.6 | 204.4 |
| Campesterol | 55.9 | 542.6 | 7.6 | 68.6 |
| β-sitosterol | 139.4 | 890.5 | 19.3 | 107.9 |

## 5. Discussion

### 5.1 Relative importance of different sources of organic carbon to Petermann Fjord

Organic carbon ($C_{org}$) in Petermann Fjord is derived from in situ marine production and input of both modern and ancient $C_{org}$ from the surrounding landmasses, which can have significant implications for the interpretation of sedimentary biomarkers. While HBI concentrations are only marginally influenced by the addition of terrestrial $C_{org}$, sterols are important constituents in both marine and terrestrial primary producers, albeit in different relative concentrations (Belicka et al., 2004; Rontani et al., 2014; Volkman, 1986, 2003; Volkman et al., 1993, 2000; Yunker et al., 1995) (Table 2). The marine sterol index can provide insights into the relative importance of either sterol group (Stein et al., 2017). This index suggests a relative dominance of terrestrial sterols throughout sedimentary unit 3 and 2 (Fig. 5E), in line with low $\delta^{13}C_{org}$ values, characteristic of terrestrial vegetation in high latitudes (-26 ‰ to -28 ‰) (Ruttenberg and Goñi, 1997) (Fig. 3B). However, while the marine sterol index remains low throughout most of unit 2, a rise in $\delta^{13}C_{org}$ indicates the increasing importance of marine organic carbon in Petermann Fjord. Sedimentary unit 3 represent grounding zone proximal sedimentation, associated with the deglacial retreat

 

of PG into the fjord (Jakobsson et al., 2018; Reilly et al., 2019). Mapped submarine landforms suggest that the retreat of PG into the fjord was rapid, driven by ice cliff instability and promoted by the retrograde slope of the outer fjord sill (Jakobsson et al., 2018). The deglacial retreat of PG might have halted at an inner sill (Tinto et al., 2015) ~30 km seaward of the present

day grounding zone (Reilly et al., 2019). This allowed for the formation of an extensive ice tongue in Petermann Fjord associated with the laminated, IRD-poor lithofacies unit 2 (Reilly et al., 2019). Thus, both units are associated with glacial erosion by PG and smaller tidewater glaciers terminating in Petermann Fjord. Where Washington and Hall Land border Petermann Fjord they are characterized by sedimentary rocks of lower Paleozoic age, composed of primarily shallow marine carbonates and evaporites (Dawes et al., 2000; Harrison et al., 2011). These contain ancient biomass that can be traced in

marine sediments. The thermal maturity of these rocks, however, indicates that fossil sterols will have been, to a large part, degraded to steranes (Parnell et al., 2007), not measured as part of this study. Thus, the increased abundance of terrestrial sterols associated with the deglacial retreat of PG is most likely related to erosion of fresh organic material, rather than input of fossil $C_{org}$ from sedimentary rocks surrounding the fjord. Campesterol and β-sitosterol are dominant in vascular plants (such as herbs), but are also found in mosses and lichens (Matsuo and Sato, 1991; Safe et al., 1975), common to the high Arctic

tundra. Since the deglacial retreat of PG occurs relatively late during the deglaciation of northern Greenland (<7,600 cal yrs BP) (Jakobsson et al., 2018), it falls into the regional Holocene Thermal Maximum (HTM) (Kaufman et al., 2004) when adjacent Washington Land was already deglaciated (Ceperley et al., 2020). Pollen records from northern Greenland and Ellesmere Island suggest a more productive terrestrial Arctic ecosystem during the HTM with herb tundra vegetation and dwarf-shrubs (Gajewski, 2015; Mode, 1996). Additionally, higher atmospheric temperatures during the HTM were most likely

associated with increased meltwater input to Petermann Fjord and drainage of the surrounding landmasses, in line with enhanced input of terrestrial sterols at this time.

Following the break-up of the deglacial Petermann ice tongue at ~6,900 cal yrs BP (unit 2/unit 1C boundary), the contribution of marine sterols to the overall sterol abundance increases, contemporaneous with an increase in the TOC concentration (Fig. 5F). A second, larger, increase in the marine sterol index is evident at 5,100 cal yrs BP (Fig. 5E), in line with decreasing XRF

Ti/Ca ratios and IRD flux (Reilly et al., 2019), suggesting decreased glacial erosion and calving activity or a decrease in the delivery of erosional products to outer Petermann Fjord. Between 5,100 cal yrs BP and 600 cal yrs BP, the marine sterol index indicates increasing dominance of marine sterols, in line with $\delta^{13}C_{org}$ suggesting primarily input of marine-derived organic matter. Compared to the deglacial ice tongue in Petermann Fjord, the re-establishment of an ice tongue during the late Holocene (<2,100 cal yrs BP) (Reilly et al., 2019) does not seem to be associated with increased input of terrestrial organic matter (Fig.

5). A possible explanation could be the lower atmospheric temperatures compared to the early Holocene (Lecavalier et al., 2017), associated with a less diverse and more sparse terrestrial flora in the high Arctic (Gajewski, 2015) and decreased meltwater input. Between 600 cal yrs BP and the top of the core the marine sterol index decreases again suggesting a relative decrease in marine organic matter input (Fig. 5E). This corresponds to the late Holocene interval with an extensive ice tongue in Petermann Fjord, intermittently covering the core site. While there is a small decrease in the TOC at this time, there is no


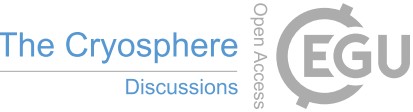





**Figure 5. Temporal changes in the environmental conditions in Petermann Fjord. From the top to the bottom: (a) Benthic foraminiferal fluxes (purple, filled in area) and benthic foraminiferal abundance in individuals per g sediment (purple line with diamonds). (b) Planktonic foraminiferal fluxes (crimson, filled in area) and plankonic foraminiferal abundance in individuals per g sediment (crimson line with diamonds). (c) Brassicasterol fluxes (light green, filled in area) and absolute concentration (light green line with dots) normalised to the amount of extracted sediment (ng g$^{-1}$ sed.). (d) IP$_{25}$ fluxes (blue, filled in area) and absolute concentration (blue line with dots) normalised to the amount of extracted sediment (ng g$^{-1}$ sed.). (e) Marine sterol index (sum of marine sterols/ sum of all sterols) (purple line). (f) TOC fluxes (green, filled in area) and TOC concentrations (wt. %, green line). (g) XRF Ti/Ca ratios (grey) and the CT >2 mm clast index (brown) (Reilly et al., 2019). (h) Reconstruction of the Holocene ice tongue extent (minimum extent in dark blue, maximum extent in light blue) in Petermann Fjord in km from the grounding zone (Reilly et al., 2019). OD1507-03TC-41GC-03PC is located 80 km from the present day grounding zone (black dashed line). Lithogenic units (1A-1C, 2, 3) in sediment core OD1507-03TC-41GC-03PC are indicated with dashed vertical lines; grounding zone proximal sedimentation (unit 3) is indicated with a grey vertical bar. The red triangles on the x-axes indicate the timing of the ice tongue break-up (~6,900 cal yrs BP) and the late Holocene inception of a small ice tongue (~2,200 cal yrs BP) (Reilly et al., 2019).**

corresponding decrease in the $\delta^{13}C_{org}$ (Fig. 3B), suggesting that marine organic carbon was still the main contributor to the TOC in outer Petermann Fjord at this time.

Sterol biomarkers, TOC, and $\delta^{13}C_{org}$ thus demonstrates that the $C_{org}$ in outer Petermann Fjord is a mixture of terrestrial and marine organic matter, linked to glacial erosion, meltwater drainage of surrounding landmasses, climate-driven vegetation changes, and marine primary productivity. However, marine organic matter was the predominant $C_{org}$ source to Petermann Fjord for large parts of the Holocene, with the dominance of terrestrial $C_{org}$ restricted to the interval of deglacial retreat of PG.

**5.2 Holocene sea-ice development in outer Petermann Fjord**

Sedimentary unit 3, a glacial diamict associated with the deglacial retreat of PG, is characterized by very low IP$_{25}$ concentrations and is almost barren of benthic and planktonic foraminifers (Fig. 5A, B), possibly suggesting reduced primary productivity under an ice shelf/thick sea-ice cover and/or high sedimentation rates close to the grounding zone diluting the concentrations of biomarkers and marine microfossils close to the margin of a retreating glacier. Subsequently, a minor increase in the concentrations of sea-ice biomarkers and benthic and planktonic foraminiferal abundances marks the onset of sedimentary unit 2 (Fig. 5A, B), in line with a marginal/sub-ice tongue regime in outer Petermann Fjord with no or only intermittent sea-ice and open water primary productivity in the fjord and low food supply to sustain benthic productivity (Jennings et al., 2020). Compared to the HBI biomarkers, the increase in sterols at the transition from sedimentary unit 3 to 2 is considerably larger, suggesting increased $C_{org}$ input to outer Petermann Fjord (Fig. 4). This is supported by an increase in TOC at the sedimentary unit boundary. The two TOC peaks in lower unit 2 represent enhanced input of terrestrial $C_{org}$, as shown by the accompanying low $\delta^{13}C_{org}$ (Fig. 3B). Overall, however, the TOC concentrations remain low throughout unit 2 with the $C_{org}$ likely being derived from a variety of sources, including low amounts of in situ produced phytoplankton, advected marine organic matter, and input of terrestrial $C_{org}$ in response to glacial erosion and enhanced meltwater drainage of surrounding landmasses.

The early Holocene ice tongue, associated with deglacial retreat of PG, collapsed around 6,900 cal yrs BP, marked by the abrupt appearance of IRD clasts in sediments across Petermann Fjord (Reilly et al., 2019) (Fig. 5G). The break-up is associated with a small increase in the sedimentary IP$_{25}$ and HBI II concentrations, as well as an increase in the benthic and planktonic



foraminiferal abundances and the marine sterol index, suggesting an increase in marine productivity (Fig. 5). Between 6,900
cal yrs BP and 5,500 cal yrs BP, $IP_{25}$ fluxes are variable but remain low, indicating reduced seasonal sea-ice concentration
during spring and low rates of sympagic productivity. A generally unstable sea-ice regime is further supported by variable
$DIP_{25}$ values (Supplementary Fig. 4). This interval is heavily influenced by the retreating PG, promoted by early Holocene
warmth. Increased meltwater runoff at this time might have contributed to unstable sea-ice conditions. Alternatively, increased
meltwater runoff might have caused low $IP_{25}$ fluxes by creating environmental conditions unfavorable for $IP_{25}$-producing
diatom species (Ribeiro et al., 2017). The simultaneous increase in TOC, benthic, and especially planktonic foraminiferal
fluxes, however, support a reduced seasonal sea-ice cover and a shift from a regime marginal to or below an ice tongue towards
ameliorated conditions with a prolonged open water season and enhanced productivity in the fjord, allowing planktonic
foraminifers to thrive (Fig. 5B). A reduced seasonal sea-ice cover is further supported by low sea-ice concentrations in Hall
Basin until at least 6,000 cal yrs BP (Jennings et al., 2011b). Regional records from northern Greenland and the western
Canadian Arctic Archipelago (CAA) demonstrate a HTM between 11,000 cal yrs BP and 5,500 cal yrs BP (Belt et al., 2010;
Briner et al., 2016; England et al., 2008; Funder et al., 2011; Jennings et al., 2011b; Kaufman et al., 2004; Knudsen et al.,
2008; Lecavalier et al., 2017; Ledu et al., 2010; Vare et al., 2009), associated with regional mean annual surface air
temperatures 3±1°C higher than pre-industrial (1750 CE) (Lecavalier et al., 2017). Thus, reduced seasonal sea-ice
concentrations in Petermann Fjord between 6,900 cal yrs BP and 5,500 cal yrs BP likely represent the late stages of the HTM
in the northern Nares Strait region.

From ca. 5,800 cal yrs BP the increase in planktonic and benthic foraminiferal fluxes steepens, associated with less variable
but still low $IP_{25}$ fluxes from 5,500 cal yrs BP (Fig. 5A, B, D). While benthic foraminifers respond to sea-ice changes primarily
through the influence on marine productivity and the resulting food supply to the seafloor (Seidenkrantz, 2013), planktonic
foraminiferal abundances in Fram Strait have been shown to be highest in the open water region and along the ice margin,
with only few individuals occurring under persistent sea ice (Carstens et al., 1997; Mayot et al., 2020). In the modern
environment of the northern Nares Strait and Petermann Fjord, planktonic foraminiferal abundances are very low in the fjord
and much higher in the mobile sea ice regime of Nares Strait (Jennings et al., 2020), characterized by a shorter seasonal sea-
ice season than outer Petermann Fjord (Fig. 2). Thus, high planktonic foraminiferal fluxes between 5,800 cal yrs BP and 3,600
cal yrs BP suggest sustained periods of seasonally open waters during summer, while the continuously low $IP_{25}$ fluxes are
consistent with reduced seasonal sea-ice concentrations and the absence of a sympagic spring bloom in outer Petermann Fjord
(Fig. 5 D). Alternatively, $IP_{25}$ production might have been suppressed due to lasting meltwater discharge into the fjord,
resulting from the influence of the retreating PG on the outer fjord environment. Both the >2 mm clast index and the XRF
Ti/Ca ratios decrease throughout this interval, suggesting a gradual reduction in the influence of glacial activity in the outer
fjord (Fig. 5G). Nonetheless, the high planktonic foraminiferal fluxes bear witness of prolonged periods of open water during
summer.

From 3,900 cal yrs BP, $IP_{25}$ fluxes increase steeply accompanied by an increase in all sterol fluxes (Fig. 4, Fig. 5D). This
suggests a shift in the ecosystem in outer Petermann Fjord, associated with a transition from a regime dominated by seasonally



open water primary productivity towards a regime characterized by MIZ conditions and enhanced sympagic productivity. Especially after 3,600 cal yrs BP, rapidly decreasing planktonic foraminiferal fluxes and steadily increasing IP$_{25}$ fluxes indicate

a progressively longer sea-ice season and enhanced sympagic productivity in outer Petermann Fjord (Fig. 5B, D). This falls into a period of long-term declining regional atmospheric temperatures recorded at Agassiz ice cap (Lecavalier et al., 2017) and in lake records in NW Greenland (Axford et al., 2019; Lasher et al., 2017). Neoglacial cooling has been observed in numerous marine and terrestrial archives around Greenland and the wider North Atlantic region (e.g. England et al., 2008; Hansen et al., 2020; Jennings et al., 2011a; Limoges et al., 2020; Vare et al., 2009), as a response to decreasing northern

hemisphere summer insolation (Marcott et al., 2013). Thus, enhanced seasonal sea-ice conditions in Petermann Fjord from 3,900 cal yrs BP (Fig. 5D), likely record the onset of neoglacial cooling in the northern Nares Strait region.

Peak IP$_{25}$ fluxes around 2,500 cal yrs BP are associated with high fluxes of marine and terrestrial sterols as well as benthic foraminifers, while planktonic foraminiferal fluxes are low (Fig. 4, Fig. 5A, B, D), indicating a prolonged seasonal sea-ice cover with only short periods of open water during summer. From 2,500-2,100 cal yrs BP a two-stepped decrease in IP$_{25}$ fluxes

is observed. Especially the second decline is accompanied by a decrease in the TOC, benthic foraminiferal flux, and a small decrease in the planktonic foraminiferal flux (Fig. 5A, B, F). A contemporaneous decline in all productivity indicators and sea-ice biomarkers, suggests a restriction in the open water and sea-ice primary productivity alike, most likely as a response to further lengthening of the sea-ice season to near-perennial sea-ice cover. This interval precedes the late Holocene inception of a small ice tongue in Petermann Fjord at 2,100-2,200 cal yrs BP (Reilly et al., 2019). After 2,100 cal yrs BP, sea-ice biomarkers,

TOC, and benthic foraminiferal fluxes recover, while the planktonic foraminiferal abundance remains low (Fig. 5A, D, F), indicating an ecosystem dominated by sympagic productivity and enhanced seasonal sea-ice cover, similar to the interval 2,500-3,600 cal yrs BP.

Around 1,300 cal yrs BP a sharp decline in IP$_{25}$, sterol, and TOC fluxes is observed, while the concentration of IP$_{25}$ decreases more gradually and the concentrations of sterols and TOC increase (Fig. 5). The sharp decreases in fluxes at this time coincides

with a large drop in the LSR (Fig. 3A), which might be biasing the flux data throughout this interval. Instead, the decline in biomarker fluxes at 950 cal yrs BP, accompanied by a decrease in the biomarker concentrations, seems to be a more reliable feature (Fig. 5). This is accompanied by declining benthic foraminiferal fluxes, and followed by decreases in the planktonic foraminiferal and TOC fluxes at 700 cal yrs BP (Fig. 5A, B, F), suggesting a return to near-perennial sea-ice conditions with reduced open water and sea-ice primary productivity, similar to the interval from 2,100-2,500 cal yrs BP. At ca. 600 cal yrs

BP a rapid extension of the Petermann ice tongue to its modern limits (Reilly et al., 2019), resulted in (at least intermittent) cover of the core site, in line with low biomarker and foraminiferal fluxes between 600 cal yrs BP and the top of the core (Fig. 5). The latter indicates low rates of primary productivity in a fjord, which is nearly completely covered by an ice tongue.

Holocene sea-ice dynamics in outer Petermann Fjord are thus marked by a period of reduced seasonal sea-ice concentrations between 6,900 cal yrs BP and 5,500 cal yrs BP, following deglacial retreat of PG and the break-up of its ice tongue (Jakobsson

et al., 2018; Reilly et al., 2019). This period corresponds to the late stages of the regional HTM and enhanced meltwater runoff from the retreating PG might have contributed to the unstable sea-ice conditions. Between 5,500 cal yrs BP and 3,600 cal yrs





BP high planktonic foraminiferal and continuously low $IP_{25}$ fluxes suggest reduced seasonal sea ice and a prolonged open water primary productivity season during summer. From 3,900 cal yrs BP increasing $IP_{25}$ fluxes, indicate progressively more enhanced sea-ice conditions and a shift to a primary productivity regime dominated by sympagic productivity from 3,600 cal

yrs BP. This suggests the onset of neoglacial cooling in the northern Nares Strait region around 3,900 cal yrs BP in line with decreasing northern hemisphere summer insolation (Marcott et al., 2013). Between 2,500 cal yrs BP and 2,100 cal yrs BP and after 950 cal yrs BP a decrease in all biomarker fluxes and productivity indicators suggests periods of near-perennial sea ice, restricting all forms of primary productivity in outer Petermann Fjord. Both the bottom (>6,900 cal yrs BP) and top (<600 cal yrs BP) of the core are influenced by an extensive ice tongue in Petermann Fjord, resulting in reduced primary productivity in

the fjord, in line with low fluxes of sea-ice biomarkers and productivity indicators throughout these intervals.

**5.3 Nares Strait sea-ice dynamics over the last 7,000 cal yrs BP**

Thus far, only one other biomarker-based sea-ice reconstruction, from station Kane2b in northwestern Kane Basin, exists in Nares Strait for comparison with our records (Fig. 1) (Georgiadis et al., 2020). Depending on the ice arch configuration in Nares Strait, Kane Basin and outer Petermann Fjord likely experience opposing sea-ice conditions during spring/early summer,

the dominant productivity season of sea-ice biomarkers. Kane2b is located near/under the southern ice arch in Nares Strait. Thus high sea-ice primary productivity and $IP_{25}$ fluxes occur during times of a stable ice arch in Smith Sound and sea-ice edge conditions exist during spring/summer (Georgiadis et al., 2020) (Fig. 6B, D). Under these conditions, sea ice in Petermann Fjord does not break up until late summer/early autumn (Fig. 2), precluding a pronounced in-ice bloom and MIZ conditions during spring/summer, resulting in relatively low sea-ice and phytoplankton biomarker concentrations (Fig. 6B, D). In years,

were only the northern ice arch forms, sea ice formed locally in Nares Strait either remains mobile throughout the winter or breaks up during early spring. Hence, no spring sea-ice/MIZ bloom is expected at station Kane2b (Fig. 6C). Instead, outer Petermann Fjord experiences spring MIZ conditions, as the formation of fast ice in Petermann Fjord is independent of the formation of fast ice in Nares Strait (Fig. 2, Supplementary Fig. 1). This is likely associated with a significant spring sea-ice bloom in outer Petermann Fjord and enhanced primary productivity related to the vicinity of the ice edge, resulting in increased

concentrations of sea-ice and primary productivity biomarkers in outer Petermann Fjord (Fig. 6C). Periods with contemporaneously low concentrations of sea-ice and primary productivity biomarkers in Kane Basin and Petermann Fjord, point towards low spring sea-ice concentration in the entire Nares Strait, likely associated with a failure of both ice arches (Fig. 6A).

In addition to the Kane Basin record, sea-ice dynamics around Ellesmere Island and NE Greenland have been inferred from

records of driftwood delivery and beach ridge formation (England et al., 2008; Funder et al., 2011), providing context with respect to Holocene sea-ice conditions in the Lincoln Sea. Driftwood is transported with Arctic multiyear ice and is deposited along the coastlines of northern Greenland and Ellesmere Island when landfast ice breaks up during summer (Funder et al., 2011). The formation of beach ridges also depends on sufficient wave action and open water along the coast. Thus, a lot of driftwood delivery to Ellesmere Island/northeastern Greenland together with abundant formation of beach ridges bears witness





of seasonally open waters along the coast (Fig. 6A). Strong sea-ice conditions in Lincoln Sea, with year-round landfast ice

along the coast, on the other hand, will result in little or no driftwood landings and reduced formation of beach ridges (Fig.

6D). Lastly, variable ice conditions in Lincoln Sea result in occasional/little driftwood landings and reduced/variable formation

of beach ridges (Fig. 6B, C).



**Figure 6. Schematic of spring sea-ice conditions and the respective sea-ice biomarker and primary productivity indicator**
**concentration patterns at OD1507-03TC-41GC-03PC and AMD14-Kane2b, as well as driftwood delivery to CMI and beach ridge**
**formation along the coast of northeastern Greenland, based on different sea-ice and ice-arch scenarios in Nares Strait. In addition**
**to the access to the coast, driftwood delivery also depends on the multiyear ice conditions in the Arctic Ocean, this is not considered**
**in this simplified schematic. (a) No ice arches, (b) Southern ice arch only, (c) Northern ice arch only, (d) Northern and southern ice**
**arch.**

While the sediment core from station Kane2b covers the last 9,000 cal yrs BP (Georgiadis et al., 2020) (Fig. 1), Petermann

Fjord did not deglaciate until ~7,600 cal yrs BP (Jakobsson et al., 2018) and the age model for OD1507-03TC-41GC-03PC is

only constrained for the last 7,000 cal yrs BP. Thus, we focus on the interval from 6,900 cal yrs BP to the present. Between

6,900 cal yrs BP and 5,500 cal yrs BP, corresponding to the late stages of the regional HTM (Kaufman et al., 2004), sea-ice

biomarker and foraminiferal fluxes in Petermann Fjord and Kane Basin (Georgiadis et al., 2020), indicate reduced seasonal





sea-ice occurrence (Fig. 7F, G). This corresponds to sparse driftwood delivery but abundant beach ridge formation around NE Greenland from 8,500-6,000 cal yrs BP, suggesting a minimum in Arctic multi-year ice and seasonally open water (Funder et al., 2011; Möller et al., 2010). Seasonally open water is further supported by maximum driftwood delivery to Disraeli Fiord and Clements Markham Inlet (CMI) (England et al., 2008) (Fig. 1, Fig. 7E). Reduced multi-year ice in the Arctic Ocean,

alongside summer coastal melt and seasonally open water around NE Greenland and Ellesmere island, is in line with reduced seasonal sea-ice occurrence in Nares Strait, suggesting predominantly mobile sea ice with no or only occasional ice arch formation and export of Arctic sea ice through Nares Strait (Fig. 8A) (Georgiadis et al., 2020).

At 5,500 cal yrs BP increasing sea-ice biomarker fluxes in Kane Basin indicate a shift towards later sea-ice retreat and ice edge productivity, interpreted to reflect recurrent formation of the southern ice arch in Smith Sound/Kane Basin between 5,500 cal

yrs BP and 3,000 cal yrs BP (Georgiadis et al., 2020) (Fig. 7F). The arrival of little auk colonies in NW Greenland at 4,400 cal yrs BP, supports the opening of the NOW lee of the Smith Sound ice arch during spring/summer (Davidson et al., 2018) (Fig. 7B). Landfast sea ice started to form in Disraeli Fiord on northern Ellesmere Island from ~5,500 cal yrs BP, covering large parts of the coast by ~3,500 cal yrs BP (England et al., 2008), and shorter periods of open water and restricted beach ridge formation occur around NE Greenland (Funder et al., 2011). Even though this suggests an intensification of landfast sea-ice in

the Lincoln Sea and the Arctic Ocean, open water along the NE Greenland coast was more abundant compared to today until at least 4,500 cal yrs BP (Funder et al., 2011). The period from 3,500-5,500 cal yrs BP, thus marks the transition from early Holocene warmth to neoglacial cooling in the wider Nares Strait region. Compared to Kane2b, biomarker fluxes do not increase significantly at 5,500 cal yrs BP in outer Petermann Fjord (Fig. 7G). Instead, benthic and planktonic foraminiferal fluxes are at their highest between 3,500 cal yrs BP and 5,500 cal yrs BP (Fig. 6A, B), interpreted to reflect reduced seasonal sea-ice

concentrations, the absence of a sympagic spring bloom and prolonged periods of seasonally open waters during summer and increased pelagic primary productivity. Alternatively, the continuous influence of the retreating PG during this interval, potentially associated with increased meltwater discharge into the fjord, might have caused low concentrations of sea-ice biomarkers in outer Petermann Fjord. Nonetheless, the high foraminiferal fluxes indicate conditions that differ from what would be expected in outer Petermann Fjord under a stable southern ice arch scenario (Fig. 6B). Possible reasons for this

include enhanced seasonality compared to the late Holocene with an early seasonal break-up of the southern ice arch and a later formation of sea ice during autumn, leading to increased annual pelagic primary productivity in Nares Strait. Thus, the spring ice edge bloom might still have occurred in the southern Nares Strait, followed by a rapid sea-ice retreat and open water conditions during summer/autumn. Thus, we suggest that this interval likely represents the transition from reduced sea-ice conditions and the lack of ice arches in Nares Strait during the late HTM towards more stable sea-ice conditions in Nares Strait

associated with the seasonal formation of a southern ice arch from ~4,400 cal yrs BP. The latter was likely associated with enhanced seasonality compared to historical records, leading to an earlier ice arch break-up and later sea-ice formation during autumn, allowing for enhanced pelagic productivity in Nares Strait.









**Figure 7. Holocene environmental changes in the wider Nares Strait region. From the top to the bottom: (a) Regional mean annual air temperature (MAAT) anomaly based on $\delta^{18}O$ at Agassiz ice cap at 25 yr resolution (Lecavalier et al., 2017) (grey line) including a $2\sigma$ uncertainty envelope (light grey filled in area) and a 5-pt running mean (red line). (b) Inferred variability in little auk numbers in seabird colonies at Annikitsoq and Qeqertaq (Davidson et al., 2018). (c) Cumulative probability distribution of calibrated $^{14}C$ ages of driftwood in NE Greenland north of 80°N (pink filled in area) (Funder et al., 2011). The numbers above the shaded area indicate the number of ages contributing to each block. (d) Cumulative probability distribution of calibrated $^{14}C$ ages of driftwood at Ward Hunt Ice Shelf and in Disraeli Fiord (yellow filled in area) (England et al., 2008). The numbers above the shaded area indicate the number of ages contributing to each block. (e) Cumulative probability distribution of calibrated 14C ages of driftwood in Clements Markham Inlet (yellow filled in area) (England et al., 2008). The numbers above the shaded area indicate the number of ages contributing to each block. (f) $IP_{25}$ fluxes (red, filled in area) and absolute concentrations (red line with diamonds) normalized to the amount of sediment extracted at AMD14-Kane2b (Georgiadis et al., 2020). (g) $IP_{25}$ fluxes (blue, filled in area) and absolute concentrations (blue line with dots) normalized to the amount of sediment extracted at OD1507-03TC-41GC-03PC. (h) Reconstruction of the Holocene ice tongue extent (minimum extent in dark blue, maximum extent in light blue) in Petermann Fjord in km from the grounding zone (Reilly et al., 2019). OD1507-03TC-41GC-03PC is located 80 km from the present day grounding zone (black dashed line). Corresponding to Fig. 8, interpretations of past ice arch dynamics in Nares Strait are marked with dashed vertical lines. The letters at the top correspond to the schematic scenarios in Fig. 6 and Fig. 8. The red triangles on the x-axes indicate the timing of the ice tongue break-up (~6,900 cal yrs BP) and the late Holocene inception of a small ice tongue (~2,200 cal yrs BP) (Reilly et al., 2019).**

From 3,900 cal yrs BP, increasing $IP_{25}$ fluxes in Petermann Fjord, attest to a progressively enhanced sea-ice season, with a shift in the primary productivity regime from predominantly pelagic to MIZ/sympagic primary productivity at 3,500 cal yrs BP. A similar increase in $IP_{25}$ concentrations from ~4,000 cal yrs BP is also observed in the western CAA, in Barrow, Victoria, and Dease Strait (Belt et al., 2010; Vare et al., 2009), in line with enhanced seasonal sea-ice formation and the onset of regional neoglacial cooling. Conversely, $IP_{25}$ concentrations in Kane Basin decrease from ~4,000 cal yrs BP until 1,400 cal yrs BP (Fig. 7F), suggesting more mobile sea ice in Nares Strait with no stable southern ice arch in Smith Sound. Instead, the driftwood delivery to CMI ceases at 3,500 cal yrs BP (England et al., 2008). This likely indicates a regime shift in Nares Strait with the northern ice arch becoming more prominent between 3,500 cal yrs BP and 1,400 cal yrs BP (Fig. 8C). In years where only the northern ice arch forms, along strait winds in Nares Strait keep locally formed sea ice mobile year-round (Vincent, 2019), creating MIZ conditions in Petermann Fjord during spring/summer (Supplementary Fig. 1), in line with high $IP_{25}$ concentrations. Between 2,500 cal yrs BP and 2,100 cal yrs BP biomarker fluxes in outer Petermann Fjord, suggest near-perennial sea-ice conditions (Fig. 7G). This is associated with some driftwood landings in CMI (England et al., 2008) and a period of no driftwood or beach ridge occurrences in NE Greenland, suggesting perennial landfast ice (Funder et al., 2011). Additionally, a small increase in $IP_{25}$ in Kane Basin (Georgiadis et al., 2020) and a little auk event at Qeqertaq (Salve Ø) at ~2,200 cal yrs BP, are in line with a stable spring/summer NOW (Davidson et al., 2018). This combination indicates a transient return to a southern ice arch dominated regime in Nares Strait from 2,100-2,500 cal yrs BP (Fig. 8B). Georgiadis et al. (2020) suggest that the period of mobile sea ice in Nares Strait between 3,500 cal yrs BP and 1,400 cal yrs BP was driven by a change in the dominant phase of the Arctic Oscillation (AO). The shift towards a positive AO between 3,000 cal yrs BP and 1,200 cal yrs BP (Darby et al., 2012) was likely associated with stronger along strait winds in Nares Strait. As ice arch formation in Nares Strait is a function of ice thickness, local wind stress, and atmospheric temperatures (Barber et al., 2001; Samelson et al., 2006), distinct atmospheric cooling spikes between ~2,500 cal yrs BP and ~1,900 cal yrs BP, recorded at Agassiz ice cap



(Lecavalier et al., 2017), might have been responsible for the enhanced sea-ice concentration and the formation of a southern ice arch in Nares Strait recorded between 2,500 cal yrs BP and 2,100 cal yrs BP (Fig. 7A).

From 1,400 cal yrs BP onwards, $IP_{25}$ fluxes in Kane Basin increase (Georgiadis et al., 2020), while they decline again in Petermann Fjord (Fig. 7F, G), suggesting a transition towards a more stable southern ice arch. These conditions are further consolidated from 950 cal yrs BP, where the sudden drop in sea-ice and primary productivity indicators in Petermann Fjord (Fig. 5, Fig. 7G) indicate near-perennial landfast sea ice in Nares Strait promoted by the recurrent formation of a southern ice arch. Driftwood is absent in CMI from 400-2,000 cal yrs BP (England et al., 2008), suggesting that the northern arch might

have been a stable feature as well (Fig. 7E, Fig. 8D). Periods of double ice arching in Nares Strait are particularly stable (Ryan and Münchow, 2017), in line with the coldest Holocene temperatures recorded at Agassiz ice cap throughout this interval (Lecavalier et al., 2017). At ~600 cal yrs BP a large jump in the extent of the late Holocene ice tongue in Petermann Fjord to its modern (pre 2010) limits, suggests a regime marginal to or under an ice tongue in outer Petermann Fjord (Reilly et al., 2019) (Fig. 7H). Thus, further interpretations of the sea-ice environment in Nares Strait are hindered due to the influence of

the ice tongue on primary productivity in Petermann Fjord.

## 5.4 Implications of sea ice for the formation and stability of the Petermann ice tongue

A recent study by Reilly et al. (2019) demonstrates that an extensive ice tongue in Petermann Fjord is a relatively recent feature compared to large parts of the Holocene. The final stage of the deglaciation in Petermann Fjord was characterized by the break-up of a deglacial ice tongue around 6,900 cal yrs BP, followed by a period of enhanced IRD and Ti/Ca ratios in the fjord

sediments (3,500-6,900 cal yrs BP; Fig. 5G), indicating increased glacial sedimentation in outer Petermann Fjord. First at ~2,200 cal yrs BP a small ice tongue re-emerged, followed by gradual growth and a rapid expansion to its modern extent around 600 cal yrs BP (Reilly et al., 2019). Thus, the mid-to-late Holocene was marked by a ~4,700 year interval where no stable ice tongue existed in Petermann Fjord (Reilly et al., 2019). The existence of an ice tongue in Petermann Fjord is determined by the interplay of several factors, including the surface mass balance of PG, the loss of glacial ice by calving, and

the basal melt rates. Local changes in sea-ice dynamics have the potential to influence both the basal melt rates and the stability of the calving front at PG. The latter depends on the formation and seasonal duration of landfast sea ice in Petermann Fjord, creating an ice mélange that stabilizes the calving front and reduces the length of the calving season (Amundson et al., 2010; Carr et al., 2015; Robel, 2017; Todd and Christoffersen, 2014). Along with the increase in $IP_{25}$ fluxes in outer Petermann Fjord, indicating a transition towards enhanced sea-ice conditions, from at least 3,500 cal yrs BP, a cessation of the IRD flux is

observed in OD1507-03TC-41GC-03PC (Fig. 5D, G) (Reilly et al., 2019). This suggests a gradual decrease in the calving of icebergs from PG during the mid-Holocene. We propose that this was partly in response to the formation of a seasonal ice mélange in the fjord, shortening the calving season. Sea ice also has the ability to influence the oceanic heat transport to Petermann Fjord, via modification to the circulation driven AW inflow by regulating the wind stress at the atmosphere/ocean interface in Nares Strait (Shroyer et al., 2017). Under modern conditions, AW inflow is enhanced at times of mobile sea ice in

Nares Strait, while landfast sea ice results in decreased oceanic heat flux to Petermann Fjord (Shroyer et al., 2017). In addition



to Nares Strait sea-ice conditions, subglacial runoff and glacial isostatic uplift of the outer fjord sill likely also influenced the inflow of AW to Petermann Fjord across the Holocene (Bennike, 2002; Cai et al., 2017; Washam et al., 2019). Recurrent formation of a southern ice arch and seasonal formation of landfast sea ice, may have persisted in Nares Strait from 3,500-5,500 cal yrs BP, 2,100-2,500 cal yrs BP, and <1,400 cal yrs BP, based on the assessment of regional sea-ice records (England

et al., 2008; Funder et al., 2011; Georgiadis et al., 2020, this study) (Fig. 8B, D). The interval from 3,500-5,500 cal yrs BP marks the transition from the HTM to neoglacial cooling in the Nares Strait region. While records from southern Nares Strait suggest an opening of the NOW and spring ice edge conditions in Kane Basin from at least 4,400 cal yrs BP, records from outer Petermann Fjord indicate low sympagic but high pelagic productivity, suggesting prolonged periods of open water during summer (Fig. 5A, B, D). Increased seasonality, with early break-up of the southern ice arch and late seasonal sea-ice formation

might be able to explain the observed patterns (section 5.3). Additionally, continued meltwater influence from the retreating PG might have affected biomarker concentrations in Petermann Fjord. Thus, this scenario likely differs from the modern conditions when a southern ice arch forms, with a longer mobile ice season in Nares Strait, which might have contributed to AW inflow to Petermann Fjord preventing the formation of an ice tongue throughout this interval. Other factors, such as increased subglacial runoff and increased surface melting of the GrIS (Vinther et al., 2009) likely contributed to inhibiting the

formation of an ice tongue in Petermann Fjord between 3,500 cla yrs BP and 5,500 cal yrs BP.

Between 2,100 cal yrs BP and 2,500 cal yrs BP regional records of sea ice (England et al., 2008; Georgiadis et al., 2020, this study) and seabird colonies (Davidson et al., 2018) suggest intermittent formation of a stable southern ice arch in Nares Strait, likely in response to regional atmospheric cooling (Lecavalier et al., 2017). This interval spans the inception of a small late Holocene ice tongue in Petermann Fjord (Fig. 7H), suggesting that increased formation of landfast ice in Nares Strait may

have reduced the oceanic heat flux to Petermann Fjord and stabilized the glacier front. Together with atmospheric cooling this likely aided the formation of a small ice tongue at ~2,200 cal yrs BP (Reilly et al., 2019). Similarly, the interval <1,400 cal yrs BP including the rapid growth of the Petermann ice tongue to its modern limits at ca. 600 cal yrs BP (Reilly et al., 2019). Regional sea-ice records indicate the onset of a more recurrent southern arch from ~1,400 cal yrs BP, with a further decrease in sea-ice and primary productivity markers in Petermann Fjord and a concurrent increase in $IP_{25}$ in Kane Basin at 950 cal yrs

BP implying the formation of more stable ice arch conditions and potentially even double ice-arching. Thus, both the inception of the ice tongue in Petermann Fjord at ~2,200 cal yrs BP and its rapid extension at 600 cal yrs BP are preceded by a transition towards stable southern ice arch conditions in the Nares Strait by 300-350 years (Fig. 7, Fig. 8B, D). More severe sea ice, with longer periods of landfast ice in Nares Strait, thus likely promoted ice tongue growth in Petermann Fjord, either directly via stabilizing the calving front or indirectly via changes to the inflow in modified AW. In light of the recent development, our

data suggest that the emerging dominance of the northern ice arch associated with year-round mobile sea ice in Nares Strait and a shorter landfast sea-ice season in Petermann Fjord, will likely contribute to the destabilization of the ice tongue. Thus, a future reduction in landfast sea ice in Nares Strait and adjacent fjords would likely contribute to enhanced mass loss from the GrIS.



**Figure 8. Simplified schematic of spring sea-ice dynamics and ice arch stability in Nares Strait including the proposed Holocene time spans that experienced the respective conditions. (A) No ice arch formation in Nares Strait resulting in year-round mobile sea ice. (B) Recurrent southern ice arch with landfast sea ice in Nares Strait and an open NOW. (C) Recurrent northern ice arch with locally formed sea ice remaining mobile in Nares Strait year-round. (D) Recurrent northern and southern ice arch with landfast sea ice in Nares Strait and a stable NOW. The location of sediment core AMD14-Kane2b is indicated with a red star, the location of OD1507-03TC-41GC-03PC is indicated with a yellow star.**

**6. Conclusions**

1. During the deglaciation (>6,900 cal yrs BP) outer Petermann Fjord experienced enhanced input of fresh terrestrial $C_{org}$, likely due to the late deglacial retreat of PG and increased glacial erosion during the early HTM, characterized by more productive Arctic tundra vegetation in northern Greenland and Ellesmere Island.



2. Following deglacial retreat of PG and break-up of the deglacial ice tongue in Petermann Fjord at 6,900 cal yrs BP, sea-ice biomarkers and productivity indicators in outer Petermann Fjord suggest reduced seasonal sea-ice occurrence and predominantly pelagic primary productivity until 5,500 cal yrs BP, corresponding to the late stages of the regional HTM. Together with other regional records, this suggests the lack of ice arches and export of sea ice from Lincoln Sea through Nares Strait.

3. The interval from 3,500-5,500 cal yrs BP marks the transition from the late stages of the HTM to the onset of neoglacial cooling. This was likely associated with a formation of a southern ice arch and the opening of the NOW from at least ~4,400 cal yrs BP. Enhanced seasonality associated with a longer open water season, caused enhanced pelagic productivity in Nares Strait and seasonal coastal melt in CMI and around NE Greenland.

4. Between 1,400 cal yrs BP and 3,500 cal yrs BP (excluding 2,100-2,500 cal yrs BP) outer Petermann Fjord was marked by enhanced pelagic productivity suggesting a marginal ice zone location during early spring/summer in response to the northern Nares Strait ice arch becoming more prominent and locally formed sea ice remaining mobile in Nares Strait year-round. A stable northern ice arch throughout this interval is supported by the cessation of driftwood delivery to CMI.

5. From 2,100-2,500 cal yrs BP, declining sea-ice biomarker fluxes alongside declining productivity indicators in outer Petermann Fjord suggest a restriction of all primary productivity, likely as a result of near-perennial sea-ice cover. In combination with a small rise in the sea-ice biomarker fluxes in Kane Basin, reappearance of driftwood at CMI, and a local abundance peak of little auk seabird colonies at Qeqertaq (Salve Ø), this suggest a transient return to a southern ice arch regime in Nares Strait. Georgiadis et al. (2020) suggest that a dominantly positive AO phase from 1,200-3,000 cal yrs BP, associated with stronger along strait winds in Nares Strait, might have physically prevented the formation of ice arches. Since ice-arch formation in Nares Strait also depends on atmospheric temperatures and sea-ice thickness, distinct atmospheric cooling spikes between 1,900 cal yrs BP and 2,500 cal yrs BP, might have exerted a positive feedback on sea-ice mechanics, counterbalancing the increased wind stress and enabling the formation of a southern ice arch.

6. The collective sea ice reconstructions in the Nares Strait and Lincoln Sea region indicate a return to a stable southern ice arch regime and potentially even double ice arching >1,400 cal yrs BP.

7. Our data demonstrates that the formation of landfast sea ice in Petermann Fjord and Nares Strait preceded major growth events of the Petermann ice tongue. This suggests that sea ice promotes an environment favorable for ice tongue growth, either directly or indirectly via stabilizing the glacier calving front and possibly decreasing the inflow of modified AW to Petermann Fjord. Conversely, a reduction in the landfast sea-ice season in Nares Strait and adjacent fjords, as observed during the last decade, will likely contribute to destabilizing local ice tongues, which might result in enhanced mass loss from the GrIS in the future.



**Data availability**

The data presented as part of this manuscript will be archived in the PANGAEA database (in progress).

**Author contribution**

The study was designed by HD and CP with help from BR and AJ. MJ led the Petermann Expedition and collected the core together with BR and AJ. HD carried out the sample processing and analyses for sea-ice biomarkers with help from MMJ. AJ carried out the processing and analyses for foraminiferal census counts. JO performed the TOC analyses. MJ, MG, and CP acquired funding for this study. HD prepared the manuscript with valuable contributions from all co-authors.

**Competing interests**

The authors declare that they have no conflict of interest.

**Acknowledgements**

We would like to thank the captain and crew, as well as the scientific party, on the Icebreaker *Oden* during *The Petermann 2015 Expedition*. Thank you to the Oregon State University Marine Geology Repository (OSU-MGR) for core archiving and sampling of the core. Further, we would like to thank Simon Belt and Lukas Smik for providing biomarker laboratory standards
and Trine Ravn-Jonsen and René Bjerregaard Madsen for laboratory support. This work was funded by the Aarhus University Research Foundation (AUFF-E-17-7-22 to Christof Pearce), the National Science Foundation Office of Polar Programs (Award 1417784 to Anne Jennings), the National Science Foundation Division of Ocean Sciences (1558679), and the Swedish Research Council (VR, 2016-04021 to Martin Jakobsson). We also thank the Swedish Polar Research Secretariat for supporting *The Petermann 2015 Expedition*. Lastly, we acknowledge the use of imagery from the NASA Worldview application
(https://worldview.earthdata.nasa.gov/), part of the NASA Earth Observing System Data and Information System (EOSDIS).

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
