# Peer review of "Holocene sea-ice dynamics in Petermann Fjord in relation to ice tongue stability and Nares Strait ice arch formation"

_The Cryosphere, 2021_

## Author Comment (AC1)

Answers to reviewer #1

Thank you very much for recognizing the importance of the data presented in this manuscript concerning the role of the marine cryosphere in stabilizing NW Greenland ice tongues. We appreciate the time and effort you have put into reviewing and improving our work. We would like to thank you for the constructive comments, which we have addressed individually below.

**Specific comments (intermediate). Numbers link to the manuscript line number.**

1. I do like the short title, but I think you could expand upon this to include the broader regional implication of the study with regards to Nares Strait sea-ice dynamics. I suggest you adapt this to indicate the wider reach of this study.

    Thank you very much for this suggestion, we have changed the title to '*Holocene sea-ice dynamics in Petermann Fjord in relation to ice tongue stability and Nares Strait ice arch formation*'. This version includes both a reference to ice tongue stability and Nares Strait ice arches, to reflect the narrative of the manuscript.

2. Figure 1a: can you add /label the deeper Atlantic water circulations on the map?

    Thank you for this comment; we have labeled the purple Arctic Atlantic Water (AAW) on the map (Fig. 1 a). Further, we are now referring to this water mass as AAW instead of modified Atlantic Water in the text, to reflect Fig. 1a.

3. Section 2: Water depth and the location of sills appears to be an important control on the AW, but it is not clear where these sills are, in particular the 220 m deep sill in Kane Basin. Can you add either a bathymetric contour or sill on your Fig 1 b? or if that looks cluttered add a little bit more detail in the text.

    We did not include the sill in Fig. 1a or Fig. 1b, as it would not have been clearly visible. Instead, we are describing in the text were the sill is located in Kane Basin ('At its shallowest, at the northern end of Kane Basin, Nares Strait is 220 m deep (Münchow and Melling, 2008)'). Thank you for this suggestion.

4. 115: can you add the current temperature of the modified AW and where it is located in the water column?

    Thank you for this comment. We have added this information in the text ('*Below (>300 m in Hall Basin), the water column is characterized by AAW (0.28-0.31 °C; (Washam et al., 2018)) that has circulated through the Arctic Ocean (Jones and Eert, 2004).*').

5. 117: in this sentence you state 'This sill impedes the throughflow of the densest AW...' do you mean it prevents AW from the south altogether or is there a density gradient in the AW? If the former I suggest you rephrase to '...impedes the throughflow of the dense AW...'

    As far as the authors are concerned, the question of AW origin in southern Nares Strait is not yet fully resolved. It is unclear to what extent AAW flowing southward through

Nares Strait spills over the 220 m shallow sill in Kane Basin. We have rephrased this section slightly to account for the ambiguity in existing observations ('*At its shallowest, at the northern end of Kane Basin, Nares Strait is 220 m deep (Münchow and Melling, 2008). This sill impedes the southward flow of AAW, suggesting that Atlantic Water at the southern end of Nares Strait is predominantly derived from the north flowing West Greenland Current (Fig. 1) instead (Melling et al., 2001).*'). Thank you for bringing this to our attention.

6. 137-140: Can you add a bit more detail for Fig. 2 in this section, such as years where landfast sea ice formation in PF acts independently to Nares Strait, and the sea-ice conditions at the time of the ice-tongue calving events?

Thank you for this comment, we have added additional information regarding the specific years we are referring to ('*The formation of landfast ice in Petermann Fjord is somewhat independent of the formation of landfast ice in Nares Strait. Landfast ice in Petermann Fjord will also form when sea ice in Nares Strait remains mobile throughout the winter (e.g. 2007, 2009, 2010; Fig. 2), although the sea-ice state in Nares Strait likely influences the timing of sea-ice break-up in Petermann Fjord in spring/summer, with earlier/later break-up during years without/with landfast sea ice in Nares Strait (Fig. 2, Supplementary Fig. 1).*').

Information regarding the sea-ice state prior to the calving events was added to the introduction instead ('*Interestingly, the 2010 calving event fell at the end of a 4-year period with no/little landfast ice in Nares Strait and associated earlier break-up of landfast ice in Petermann Fjord (Fig. 2). The smaller calving event in 2012, on the other hand, followed the re-establishment of extensive landfast ice in Nares Strait in 2011 (Fig. 2).*').

7. 140: add reference and detail on how productivity changes with sea-ice conditions. Also, how does Supplementary Figure 1 show PF with ice edge conditions in spring/summer? Can you add an example year in the text in line 142 '…during years (e.g., ).

As of yet, there are no studies on the importance of the sea-ice edge position for the local primary productivity in Nares Strait. The assumptions we are making here are our own, based on the observed sea-ice dynamics from satellite images of the last decades and the positive correlation of marine sterols with sea-ice biomarkers throughout the presented record, indicating the importance of sea ice for the local primary productivity. We have rephrased this section, to make sure the reader is made aware that these are our suggestions of how sea ice changes the primary productivity regime ('*Additionally, we propose that the sea-ice dynamics in Nares Strait likely have important implications for the primary productivity regime in Petermann Fjord, with ice edge conditions in spring/ early summer in years with mobile sea ice in Nares Strait (e.g. 2007 and 2009; Supplementary Fig. 1), while under landfast ice conditions in Nares Strait the spring/ early summer ice edge is situated several hundred kilometres to the southwest in Kane*

*Basin/Smith Sound (e.g. 2013 and 2014; Supplementary Fig. 1). Since the spring ice edge is a highly productive system, especially important for the sympagic algal bloom (Ardyna and Arrigo, 2020; Leu et al., 2015; Wassmann and Reigstad, 2011), the position of the Nares Strait ice edge is likely to play a role for the local primary productivity.'*).

8. Section 3.1, pg 7: I would like to see a bit more detail for the core description added in this section. In line 179 and 180, can you add more detail on the '…coarser particles. ' and '...coarse material…' respectively, such as rough grain size? Also, can you add where the core is laminated in this section rather than in the results?

> Thank you very much for this comments, we have added additional information on the laminations and coarse particles and their origin in the different sedimentary units ('*Sedimentary unit 3 (ca. 518-555 cm) is a massive diamict composed of a sandy mud with abundant coarse clasts and XRF Ti/Ca ratios around 0.05 (Reilly et al., 2019). The clasts found in this unit, likely do not represent ice rafted debris (IRD), but instead are related to proximity of the grounding zone (Reilly et al., 2019). Unit 2, a clayey laminated mud with no or very low concentrations of coarser material (IRD), is found between ca. 398-518 cm (Reilly et al., 2019). (…) Subunit 1C is marked by a decreasing trend from high to intermediate IRD occurrence, with intermediate IRD abundances continuing during subunit 1B, followed by low IRD abundance during subunit 1A (Reilly et al., 2019).*').

9. Section 3.3 Sea-ice biomarker methodology is very detailed. Does all this information need to be in the main text? Can some of it be moved into the supplementary section e.g., Table 2?

> We are not sure if the reviewer refers to section 3.3.2 were the lipid biomarker extraction and analysis is described, or section 3.2 Sea-ice biomarker methodology. If the former, the analyses of the other proxies are also described in detail, which is why we do not think it would be appropriate to move only this information to the supplementary material. If the latter, we felt it was necessary to give an overview of the methodology and its caveats (e.g. in fjord environments (Ribeiro et al. 2017)) and regarding the difficulties measuring HBI III in Petermann Fjord. Even though sterols are commonly used in sea-ice reconstructions, we felt it was necessary to discuss their caveats and establish a framework for how we intend to interpret them in this study (and with what restrictions).

10. Section 3.3: Foraminifera abundances are interesting and provide potential information on productivity, but assemblage data would also be very insightful with regards to the sea-ice or ice tongue presence in the PF as well as the presence of AW (e.g., presence of the foraminifera Cassidulina neoteritis). Is there a reason assemblage counts have not been included here?

> Thank you very much for this comment. Foraminiferal assemblage data was not included in this study, as it will be published in a separate manuscript.

11. 366-375: this section describes the DIP25 results, but this is in the Supp info. Is there a reason for this or can this data set be moved into the main text?

> Thank you for this suggestion. Considering the uncertainty of the $DIP_{25}$ index, it does not contribute to the main narrative of the manuscript. Instead, we only refer to more variable/less variable $DIP_{25}$ values in the discussion (so we are making very limited use of this index). This is why the graph is not included in the main text but rather the supplementary material.

12. General results section comment. The core stratigraphy and environmental interpretations are sporadically included within the results section. In doing so there is some repetition e.g., line 332 and 349 and new information on the core is added intermittently (e.g., lines 353-354). I would suggest removing the interpretation and sedimentology detail out of the results section unless it is needed to justify why data has not been included (e.g., lines 329-331). Instead, I would like to see this detail added to the methods section (3.1) when the core is initially described and within the discussion section where the multiproxy results can be linked to the core interpretation.

> Thank you for this comment. We agree with the reviewer that there was some repetition and interpretations in the results section. We have removed these and moved the information regarding sedimentology to section 3.1 instead (see answer to comment #8).

13. Lines 470-475: could another explanation for the lower terrestrial input during the reestablishment of the ice tongue be related to grounding-line proximity to the core location?

> During this time period the grounding zone is further away from the core site compared to the mid/early Holocene. As the reviewer points out, this can result in decreased delivery of terrestrial $C_{org}$ transported for example in the meltwater discharged at the grounding zone. We have added this to the manuscript ('*Compared to the deglacial PG ice tongue, the late Holocene ice tongue (<2,100 cal yrs BP) (Reilly et al., 2019) does not seem to be associated with increased input of terrestrial organic matter to outer Petermann Fjord (Fig. 5). A possible explanation could be lower atmospheric temperatures compared to the early Holocene (Lecavalier et al., 2017), associated with a less diverse and more sparse terrestrial flora in the high Arctic (Gajewski, 2015) and decreased meltwater input. Another reason could be increased distance of OD1507-03TC-41GC-03PC from the PG grounding zone during the late Holocene (Reilly et al., 2019), resulting in reduced delivery of meltwater-derived $C_{org}$.*').

14. Figure 5h: the reconstructed ice-tongue extent (km) at ~590-600 cm doesn't seem to match that produced in Figure 5 of Reilley et al (2019). I also suggest 5h extends back to 555 cm to match the datasets for this study.

Thank you very much for noticing this. We have adjusted Figure 5 accordingly. The depth axis at the bottom is now only covering 560 cm, to reflect the depth interval analysed as part of this study.

15. 518: The author state that increased meltwater runoff may have led to sea-ice instability. Can they add a bit more detail on the process and/or reference?

Thank you for this comment. For clarity, we have removed this sentence from the revised manuscript. In the original manuscript, this was supposed to refer to periodic meltwater discharge, which might affect sea-ice conditions via its effects on salinity and the freezing point of surface waters in Petermann Fjord.

16. Section 5.3, line 593-595: an assumption made is that dependent on ice arch configuration PF and Kane Basin will have opposing sea-ice conditions. What is this assumption based upon? The 4 satellite images in Supplementary figure 1 or published observations? If the latter then can you add the reference to the text? If this is based on the authors' observations, it would be worth adding a figure like Fig 2 that includes Kane2b sea-ice conditions to show this modern relationship and that there is indeed opposing sea-ice conditions to strengthen this assumption. This could be added to the Supp info.

Thank you for this comment. This is our own assumption, we propose in this manuscript that based on the ice arch configuration in Nares Strait the position of the spring sea-ice edge moves relative to the two core sites in Kane Basin and outer Petermann Fjord. We have added additional explanation to the text (see answer to comment #7). During years with a stable southern ice arch, the spring/summer ice edge and associated spring MIZ bloom of biomarker relevant sympagic diatoms is closer to Kane2b, and vice versa during years with no ice arch or only a northern ice arch, where a local ice edge forms at the mouth of Petermann Fjord.

In years when a southern ice arch forms, the break-up of landfast sea ice in Kane Basin and Hall Basin (indicated in Fig. 2) is almost synchronous with only few days' difference. Thus, adding the day of year of ice break-up in Kane Basin to Figure 2 would not add much additional information. Instead, we have added the day of year when the southern ice arch broke up (according to Vincent (2019); see below).

[Figure]

Revised Figure 2. Modern ice arch dynamics in Nares Strait. (a) Annual mean ice area flux through southern Lincoln Sea flux gate (2000-2009 (Kwok et al., 2010), 2017-2019 (Moore et al., 2021)). (b) Difference between ice break-up in Hall Basin and Petermann Fjord in days. After 2010/2012 the differences increase due to the larger sea-ice area in Petermann Fjord following the retreat of the ice tongue. (c) Time series of approximate landfast sea-ice break-up in Petermann Fjord (light blue) and Hall Basin (dark blue) from 2000-2020 estimated from https://worldview.earthdata.nasa.gov/. The orange triangles represent the day of year when the southern ice arch collapsed (Vincent, 2019). Where the Hall Basin and ice arch break-up record have a day of year of zero, landfast sea ice did not form in Nares Strait throughout the entire year. Where cloud cover inhibited the exact determination of the day of landfast sea-ice break-up, the average between the last clear day with landfast ice and the first clear day following ice break-up was used (Hall Basin: 2000,2016; Petermann Fjord: 2001, 2004, 2005, 2013, 2017, 2018). Error bars indicate the timespan where cloud cover inhibited clear determination of the sea-ice state. The dashed vertical lines indicate years of substantial ice tongue calving in Petermann Fjord.

17. 640: This is the first time that 'The arrival of little auk colonies…' have been mentioned. Can you add their significance with regards to understanding sea-ice conditions?

Thank you for noticing that information on little auk ecology was missing. We have added additional information on the feeding habits of little auk and their importance in understanding NOW dynamics. Additionally, we have added recently published evidence that supports strong NOW conditions from ca. 4,400 cal yrs BP ('*Little auk are zooplanktivore, feeding on the abundant copepod population of the NOW. Thus, large colonies of little auk in Greenland are only found in vicinity of productive polynyas, where open water is available for foraging upon their arrival in spring (Davidson et al., 2018). Productive and strong NOW conditions from 4,400 cal yrs BP are also inferred based on foraminiferal assemblages from the central polynya region (Jackson et al., 2021).*').

18. 654-662: Enhanced seasonality is an interesting outcome. Are there any other records (marine/terrestrial) that also indicates enhance seasonality to support this interpretation? Why would seasonality be enhanced during this period? You also start the sentence 'Possible reasons…' what other potential reasons are there for this and why is seasonality your preference?

Thank you very much for this comment. We have re-written this section to incorporate the possible reason for enhanced seasonality during this time period and we re-phrased it to 'A possible reason…' as we cannot think of any other reasons at this point that could cause the observed proxy patterns: '*A possible reason for this could be enhanced seasonality, in particular enhanced winter cooling, due to the increasing sea-ice extent in the Arctic Ocean (England et al., 2008; Funder et al., 2011), while summer insolation was still relatively high (though decreasing) compared to the late Holocene (Fig. 7). Modelling studies have shown that the loss of sea ice in the Arctic Ocean during the early Holocene counteracted the increased seasonality prescribed by insolation forcing, due to enhanced ocean-atmosphere heat flux during winter (Fischer and Jungclaus, 2011). The Arctic Ocean thus acted as a heat reservoir with increased latent and sensible heat flux during winter as a result of the reduced sea-ice cover (Fischer and Jungclaus, 2011). Conversely, decreasing summer insolation and the associated increase in Arctic Ocean sea-ice cover during the mid-Holocene (Fig. 7) strengthened the insulating effect of sea ice on the ocean and led to pronounced cooling during autumn/winter (Fischer and Jungclaus, 2011). Enhanced seasonality could explain the observed proxy patterns in Nares Strait between 5,500 cal yrs BP and 3,500 cal yrs BP, with an early seasonal break-up of the southern ice arch, leading to increased pelagic primary productivity in northern Nares Strait. This way, the spring ice edge bloom might still have occurred in the southern Nares Strait, followed by a rapid sea-ice retreat and open water conditions during summer/autumn, as recorded in outer Petermann Fjord. Thus, we suggest that this interval represents the transition from reduced sea-ice conditions and the lack of ice arches in Nares Strait during the late HTM towards more stable sea-ice conditions associated with the seasonal formation of a recurrent southern ice arch from at least ~4,400 cal yrs BP.*'

19. 661: What historical records are you referring to?

In the original manuscript, we intended to refer to the modern sea-ice dynamics of Nares Strait, but realize the wording was confusing. Thus, we have deleted the particular sentence.

20. 731-732: Here the authors proposed the formation of seasonal ice melange in the fjord shortened the calving season. What is the evidence of this? Can you refer to the appropriate results?

> Thank you for this comment. We have changed the wording from ice mélange to seasonal landfast sea ice, as we cannot say for sure (even though it is likely) that an ice mélange formed.

21. 741-743-: I found the sentence starting 'Thus, this scenario…' hard to follow. Can you rephrase this?

> Thank you for this suggestion. We have reworded this sentence and split it up in two (*'Thus, unlike today, the formation of a southern ice arch between 5,500 cal yrs BP and 3,500 cal yrs BP did likely not lead to a prolonged landfast ice season in Nares Strait. Consequently, the longer mobile sea-ice season in Nares Strait might have contributed to AAW inflow to Petermann Fjord preventing the formation of an ice tongue throughout this interval.'*)

**Specific comments (technical).**

Thank you for these comments. We have adjusted the text and figures accordingly. However, we did not add a reference to the sea-ice categories (comment 27), as this is supposed to be a clarification of the different descriptors we are using throughout the manuscript for consistency when talking about relative changes in the sea-ice concentrations. We have clarified this in the manuscript text (*'For clarity, we will describe past sea-ice conditions using the following categories in order of increasing average sea-ice concentration: ice free, reduced seasonal sea ice, enhanced seasonal sea ice, and near-perennial (in order of increasing average sea-ice concentration for a given area).'*).

22. 41: add a, after Historically.
23. Figure 1A: there are purple currents that have not been labelled. Can you state what these are in the caption?
24. 108: remove 'the' before Baffin Bay
25. 109: capital C for Robeson Channel
26. 134: remove 'vast'
27. 195-197: add ref to end of the sentence
28. Section 3.3 Planktonic and benthic foraminiferal abundances: change to 3.4
29. 298 and 300: add ref to end of sentences.
30. Line 391: change to 'These become especially apparent in the dinosterol and campesterol concentrations between 400-500 cm and the $\beta$-sitosterol concentrations (add core depth)
31. 447: Add Washington and Hall Land onto Figure 1c
32. 544: I suggest you change the wording '…bear witness of…' to suggest prolonged
33. 560: I suggest you do not start the sentence 'Especially…'. Also, it is not clear from this sentence whether both declines in IP25 fluxes are accompanied by a decrease in TOC but the second is more significant, or only the second IP25 decline is accompanied by a decrease in TOC.
34. 602 and 603: be consistent with landfast or fast.

35. 613- 615: I suggest you rephrase this sentence to 'Thus, abundant driftwood delivery to Ellesmere Island/northeastern Greenland together with the formation of beach ridges is indicative of seasonally open waters along the coast (Fig. 6A).

36. Figure 8: A simplified version of Fig. 6 is not needed. I suggest you remove Fig. 8 and add the ages to each panel in Fig. 6

37. Supplementary Figure 1: can you include what ice arch state each image is showing in the caption and label each satellite image a), b), c) and d)?

38. Figures: overall the figures are nicely presented. My only minor criticism is that the y-axis does not line up to the graphs. Whilst I appreciate this is done to enable the graphs to fit on one page, it does make it harder to see the values and which axis relates to which graph. In some cases, the lower axis overlaps with the overlying graph and the overlying graph is covering some of the data points e.g., Fig 4e. Can the authors find a way to make this easier for the reader to view?

---

## Author Comment (AC2)

Answers to reviewer #2

Thank you very much for the time and effort you have put into reviewing and improving our work. We would like to thank you for your constructive comments, which we have addressed individually below.

**General comments:**

- The paper is lengthy and unnecessary repetitions between sections should be avoided (e.g., Introduction and Regional settings).

  Thank you very much for this comment. We have carefully screened the manuscript for repetition and have removed those sections. Further, in accordance with comments by reviewer #3 we have combined section 5.1 and 5.2, where a lot of repetition occurred.

- Neoglacial should be capitalized

  Thank you for noticing this. We have changed this throughout the manuscript.

  **Figure 1**.

- Please precise what the extent of the NOW represents (average between year-a/year-b, and season). It is not necessary to repeat the same sentence twice "The approximate extent of the NOW is indicated with a black dashed line." The red box already indicates the close up.

  Thank you for this comment. We have added the missing information to the caption of Figure 1 and removed the repetition ('*The approximate extent of the NOW polynya (mean extent in May 1954-1968) (Dunbar, 1969) is indicated with a black dashed line.*')

- From the caption, one may interpret that core AMD15-Kane2b was analysed as part of this study. Should add reference to paper.

  This was not our intention. We have added the reference Georgiadis et al. (2020) to the figure caption, to clarify that Kane2b is not part of the new work presented in this manuscript.

  **Line 107**. Since this sentence is already the first of the introduction, it should be left out here. Avoiding repetition will help reduce the length of the paper.

  We have removed this sentence while screening the manuscript for repetition.

  **Lines 150-152**. This sentence does not read easily. Please try to reformulate.

  Thank you for bringing this to our attention, we have rephrased the sentence to '*Modelling studies suggest that the displacement of water masses in Nares Strait in response to the prevailing sea-ice regime also affects Petermann Fjord, with enhanced inflow of warmer, saltier AAW during times of mobile sea-ice (Shroyer et al., 2017).*'.

**Line 205.** Should also mention HBI II-producing species. Co-production implies same species, strictly? Please clarify.

> Thank you for this comment. We have added the known species producing HBI II to the text ('*Given its co-production in H. spicula, H. kjellmanii, and Pleurosigma stuxbergii var. rhomboides (Brown et al., 2014), HBI II co-varies with $IP_{25}$ in the Arctic realm. Thus, the typically higher HBI II concentrations can provide additional information during times of low/absent sedimentary $IP_{25}$.*').

**Figure 2**.

- Line 166 should be Hal**l**
- "After 2012/2012, the differences increase due (…)" remove "s" in "increase".

> Thank you for both comments, we have corrected the typos.

**Lines 222-225**. This information does not seem necessary since both sterols are not direct indicators of sea ice conditions. Could only keep the last sentence of this paragraph.

> We have removed the first sentence of this section, but kept '*Depending on the predominant primary producers, high concentrations of brassicasterol and dinosterol are reported from ice-free regions, the MIZ, and under varying concentrations of seasonal sea ice (Xiao et al., 2015). In general sterol concentrations are low under perennial sea ice in the Arctic Ocean, where all primary productivity is impeded due to limited light availability (Xiao et al., 2015).*'. This is to provide the reader with a brief overview of how sterols (via general changes to the primary productivity) are affected by different sea-ice regimes.

**Lines 234-236**. These sentences seem out of place. This information (storage of research material and subsampling) should be included in the first paragraph of the Method section.

> Thank you for this comment, we agree and have moved this information to the method section.

**Line 252**. "The $d^{13}C_{org}$ reproducibility **of** non-acid pretreated (…)" add "of"

> We have corrected this typo, thank you for noticing.

**Line 284**. "Dry bulk densities were calculated (…)" this is repetitive with Line 258. No need to repeat multiple times. If applies to all, could be included in the first paragraph of the method section.

> Thank you, we have removed the repetition and added the information to the first paragraph of the method section, as the reviewer suggests ('*The dried samples were weighed and dry bulk densities (DBD) were determined from the samples respective volume and dry weight.*').

**Line 288**. Could rephrase to "A multitude of environmental factors determine the abundance and species composition of benthic and planktonic foraminifer assemblages"

> We have rephrased this sentence, as the reviewer suggests.

**Line 294**. Replace light conditions by light availability.

Thank you for this comment, we have changed this in the manuscript.

**Results section**

- Verify significant digits for the foraminifer fluxes.

    In the revised version of the manuscript, we have rounded the foraminiferal fluxes to the nearest whole number.

- Why aren't the results reported against the age rather than depth (or both)? This would make it easier to follow the changes/story in the context of Holocene climate variability.

    The results are reported against depth for two reasons; 1) To show the close connection with the lithology of the core (i.e. sedimentary units). 2) Because the age model does not cover the entire depth interval studied throughout this core. Thus, we have kept the results reported against depth in the revised manuscript, while the data is reported against age in the discussion.

- The same structure (i.e., order) could be maintained between the Method and Result sections.

    Thank you for this comments, we have rearranged the results section accordingly in the revised manuscript.

**Line 317**. Add "s" to "unit"; same in **Line 342** (please correct throughout)

We have corrected this throughout the manuscript, thank you for noticing this.

**Figure 3**.

- Should include an age axis.

    As discussed above, we have kept the description of results against depth. Thus, we have not added an age axis to figure 3.

- Remove "s" in foraminifer**s** results
- Turquois**e**

    Thank you we have corrected both typos.

- Should mention in the captions why the fluxes do not extend beyond 400 cm.

    Thank you for this comment. We have added the following statement to the figure caption: '*Fluxes do not extend beyond ~400 cm, as this depth corresponds to the lowermost available radiocarbon date (Reilly et al., 2019).*'.

- Should include HBI concentrations normalized to TOC.

    Our interpretations in the discussion section are not based on the HBI concentrations normalized to TOC; therefore, we have decided to keep those in the supplementary material, to not overcrowd Figure 3.

**Line 441**. "Sedimentary unit 3 represent**s**" add a "s"

    We have corrected the typo, thank you for noticing.

**Lines 447-449**. This sentence is not clear.

    Thank you for this comment. In accordance with a comment by reviewer #3 we have combined section 5.1 and 5.2 of the discussion and deleted this particular sentence in the process.

**Figure 5.** The title of this figure is "Temporal changes in the environmental conditions in Petermann Fjord". Yet, there is no indication of "environmental conditions" in this figure, with the only exception of the ice tongue length from the grounding zone. To make this figure truly distinct from the other figures presented in the results, more interpretive information could be added.

    We are not entirely sure what the reviewer would like us to add to the figure. It is termed 'temporal environmental changes' because an age axis is included for the first time in the manuscript. We would like to refrain from adding interpretative information such as arrows to indicate more/less sea ice, as we want to allow the readers to gain an independent understanding of the data.

**Line 600.** "In years **when** only the northern (…)" replace were by when.

    We have corrected the typo, thank you for noticing.

What about pelagic productivity?

    In this study we focus on the sea-ice biomarkers and associated phytoplankton biomarkers presented at Kane2b ($IP_{25}$ and HBI III) and at OD15-03TC-41GC-03PC ($IP_{25}$, sterols). The schematic and its description thus focus on the dynamics of these biomarker groups in response to the predominant position of the spring sea-ice edge. However, we have revised Figure 6 (see below) to account for the difference in phytoplankton biomarker used in this study and in Georgiadis et al. (2020).

**Figure 6**. Great representation of scenarios.

    Thank you very much for this comment.

**Lines 695-704.** This section as well as the interpretation of the southern/northern ice arch conditions (e.g., Line 738), need to be revisited. The $IP_{25}$ fluxes reported by Georgiadis et al. 2020 are very low around 2,200 cal yrs BP. Authors from this paper indicate that $IP_{25}$ fluxes are at their lowest between 2.2 and 1.1 cal ka BP, which they interpret to result from an unstable southern ice arch in Kane Basin from 3.0 cal years BP. Little auks are present in low numbers at lake Annikitsoq and only one discrete event of

local peak in bird abundance is reported from lake Qeqertaq (Davidson et al. 2018). These data do not point towards a stable spring/summer North Water polynya and return to a southern ice arch dominated regime in Nares Strait, as suggested here.

**Lines 760-762**. See comment above. Perhaps supported for the rapid extension at 600 cal yrs BP and double-arching, but the evidence from Kane Basin is pointing towards less stable southern ice arch from 3.0 to 1.1 cal yrs BP.

Thank you very much for these comments. We agree with the reviewer that this time period experienced relatively low $IP_{25}$ fluxes in Kane Basin. Nevertheless, a small increase can be observed between ca. 3,000 cal yrs BP and 1,400 cal yrs BP (Rev. Fig. 1). This increase alone is not convincing, however it is associated with a decrease in HBI III fluxes at Kane2b (Rev. Fig. 1). Georgiadis et al. (2020) interpret HBI III to represent ice-laden and fresh surface waters in Nares Strait. This can be a result of either the break-up of the southern ice arch (variable HBI III fluxes depending on the seasonal timing of break-up) or year-round mobile sea ice in Nares Strait (northern ice arch or no ice arch scenario). Thus, a small increase in $IP_{25}$ and a decrease in the HBI III fluxes points towards intermittent formation of a southern ice arch and a longer landfast ice season in Kane Basin. Our interpretations are further supported by a recently published study of foraminiferal assemblages in the central NOW region (Jackson et al. 2021), suggesting strong polynya conditions and $CO_2$-rich brine formation throughout this interval causing dissolution of calcareous foraminiferal tests but preservation of agglutinated species (Rev. Fig. 1). The foraminiferal assemblage interpretation is supported by measurements of total Sulphur, which decrease throughout this interval, in line with enhanced bottom water ventilation as a result of brine formation (Jackson et al. 2021) (Rev. Fig. 1).

[Figure]

**Review Figure 1.** Evidence for southern ice arch formation between 2,500 cla yrs BP and 2,100 cal yrs BP from the wider Nares Strait region. From the bottom to the top: Petermann Glacier ice tongue extent (Reilly et al., 2019), outer Petermann Fjord IP$_{25}$ concentration (blue line) and fluxes (blue filled area), Kane Basin IP$_{25}$ concentration (red line) and fluxes (red filled area) and HBI III concentration (green line) and fluxes (green filled area) (Georgiadis et al., 2020), Central NOW agglutinated (deep purple) and calcareous (light purple) foraminiferal abundances and total Sulphur (TS) (Jackson et al., 2021), driftwood occurrence at Clements Markham Inlet (England et al., 2008), Little auk occurrence at Qeqertaq (Salve Ø) (Davidson et al., 2018), regional mean annual air temperature (MAAT) anomaly based on $\delta^{18}$O at Agassiz ice cap at 25 yr resolution (Lecavalier et al., 2017).

These considerations led us to revise the schematic Fig. 6 to account for the difference in phytoplankton biomarker used at Kane2b and OD15-03TC-41GC-03PC (Revised Fig. 6). Georgiadis et al. (2019) measured HBI III as phytoplankton biomarker, while this study from outer Petermann Fjord measured sterols as phytoplankton biomarkers. HBI III is interpreted to reflect ice-laden surface waters, which can occur during times of mobile sea ice and after the break-up pf the southern ice arch (Georgiadis et al., 2019). For the latter, HBI III fluxes likely depend on the seasonal timing of ice arch break up, thus, the southern ice arch and double ice arch scenario can be associated with variable HBI III concentrations, while no ice arches and a northern ice arch only are likely associated with moderately high HBI III concentrations in Kane Basin. In outer Petermann Fjord marine sterols are significantly positively correlated with $IP_{25}$, suggesting a close relationship between sterol-producing pelagic productivity and sympagic productivity. Thus, their concentrations are likely to co-vary in response to the prevailing sea-ice state.

[Figure]

**Revised Figure 6.** Schematic of spring sea-ice conditions and the respective sea-ice biomarker and primary productivity (HBI III at Kane2b, sterols at OD1507-03TC-41GC-03PC) indicator concentration patterns at OD1507-03TC-41GC-03PC (yellow star) and AMD14-Kane2b (red star), as well as driftwood delivery to CMI and beach ridge formation along the coast of north-eastern Greenland, based on different sea-ice and ice-arch scenarios in Nares Strait. In addition to the access to the coast, driftwood delivery also depends on the multiyear ice conditions in the Arctic Ocean, this is not considered in this simplified schematic. (a) No ice arch formation in Nares Strait resulting in year-round mobile sea ice, (b) Recurrent southern ice arch with landfast sea ice in Nares Strait and an open NOW, (c) Recurrent northern ice arch with locally formed sea ice remaining mobile in Nares Strait year-round, (d) Recurrent northern and southern ice arch with landfast sea ice in Nares Strait and a stable NOW. The time periods in the top left corner of each panel correspond to the proposed Holocene interval that experienced the respective conditions.

---

## Author Comment (AC3)

Answers to reviewer #3

Thank you very much for recognizing the importance of our study and for the time and effort you have put into reviewing and improving our work. We would also like to thank you for the constructive comments, which we have addressed individually below.

1. There is a very clear and detailed introduction providing background context for the study. I think it would be useful to include on fig 1 other cores in Petermann Fjord that have been used in the Reilly et al (2019) paper to assess the development of the ice tongue.

   > Thank you for this comment. We have added the additional cores discussed in Reilly et al. (2019) to Figure 1 of this manuscript.

2. Methods provide a detailed rationale for the proxies used – very clearly outlined and justified. The datasets are then very clearly described in the results section. There are numerous well developed figures used to illustrate the data. One aspect that it would be useful to consider adding is a plot showing the development of the age model – even though this may have been published elsewhere, I think it would be useful context here.

   > Thank you for this suggestion. We describe the development of the age model in quite some detail. Thus, we do not think a figure is necessary (as there are already quite a few figures in the manuscript). In this case, the reader is referred to Reilly et al. (2019) for a more detailed insight to the age model.

3. The data are plotted against depth in many of the figures (3, 4 and 5) – this is fine for the initial results description, but I think it would be clearer if the data were then plotted against age to help in the discussion section. Fig 5 shows an age axis, but it is not always easy to follow.

   > Thank you for this comment. We have switched around the position of the age axis and depth axis in Figure 5 to make it easier to interpret the data against age.

**Discussion**

4. I found Section 5.1 confusing in places - it seems to assume an interpretation that is more clearly presented in section 5.2 – and often based on the earlier Reilly et al. 2019 paper. For example in lines 463 – 464: 'Following the break-up of the deglacial Petermann ice tongue at ~6,900 cal yrs BP (unit 2/unit 1C boundary)' ok, this does seem sensible, but would be good to be clear what the evidence is for the break-up of the ice tongue based on the data presented here (based on IRD input, disappearance of laminations, increase in marine productivity?).

   > Thank you for this comment. The ice tongue dynamics are published in Reilly et al. (2019) based on sedimentary proxies from multiple cores inside the fjord. While our data support the timing of ice tongue changes, the aim of this study is not to determine ice tongue dynamics, which is why we rely on the timing published in Reilly et al. (2019) for this. Thus, we take the timing of ice tongue break-up/re-establishment as given. However, we discuss our data in light of ice tongue dynamics and how it supports the interpretations made in Reilly et al. (2019).

5. Development of the ice tongue (Lines 468 – 490) - It is not clear what the evidence is for the development of the ice tongue again from 2,100 cal BP (transition from unit 1C to 1B). Again, this is partly based on Reilly et al 2019 and supported by data presented in this paper, but this discussion comes in a later section. The decrease in marine productivity from 600 cal BP seems clear support for the development of the ice tongue…

> Thank you for this comment, see answers to comments 6 and 7.

6. I wonder if section 5.1 is actually needed as a separate section? I think it would actually be clearer if this section was removed and key parts subsumed with the rest of the discussion section. This would likely save on repetition and make the paper easier to follow.

> Thank you for these comments. We have combined section 5.1 and 5.2 to avoid repetition and to bundle our interpretations of with previous studies (e.g. Reilly et al. 2019) more clearly.

7. Section 5.2 presents the interpretation of the dataset very clearly and would actually be better coming before section 5.1 (or as suggested above remove section 5.1). Perhaps be clearer what the evidence is for initiation of ice tongue development from 2100 cal BP – based on Reilly et al and other cores closer to the grounding line (ie up-fjord)?

> Thank you for these comments. We have added additional information regarding the evidence presented in Reilly et al. (2019) for the inception of the ice tongue at 2,100-2,200 cal yrs BP ('*This interval precedes the late Holocene inception of a small ice tongue in Petermann Fjord at 2,200-2,100 cal yrs BP, inferred from Ti/Ca ratios and the stacked >2 mm clast index from four cores in Petermann Fjord (Reilly et al., 2019).*').

8. The discussion sections are very detailed, and from what I can tell they seem to be well supported by the datasets presented here. I think there is probably scope for reducing the length of the discussion (avoiding repetition). I think having the key datasets presented together plotted against age rather than depth would also help the reader. Fig 7 presents some of the data form the paper plotted against age, alongside a range of other datasets – this is a really useful to visualise the various datasets, but having more of the primary data presented in this paper plotted against the age model would be helpful.

> Thank you for this comment. In accordance with combining section 5.1 and 5.2 and comments by reviewer #2 we have carefully screened the discussion for repetition to reduce the length of the manuscript. As mentioned above we have changed the position of the age and depth axis in Fig. 5 to make visual comparison of the data against age easier for the reader. However, we kept the depth axis in Fig.5 as the age model for core OD15-03TC-41GC-03PC does not cover the entire depth range analysed as part of this study.

---

## Editor Decision (ED1)

**Editor comments on Detlef et al. resubmission**

I would like to thank the authors for submitting a thorough response to the reviewers' comments and a revised version of their manuscript, and the reviewers for confirming that their comments have been appropriately addressed.

Two of the reviewers raise minor technical issues that should be addressed prior to publication. In addition, I list a number of suggestions below that are intended to improve the clarity of the text. In general, the article is very well written, and I leave it to the authors to decide whether to adopt my very minor suggestions.

This is an important and comprehensive multi-proxy study into Holocene sea ice conditions in Petermann Fjord and the wider implications of Nares Strait ice arch formation. I am delighted to recommend this article for publication once the minor technical issues documented below and in the reviewer reports have been addressed.

Thank you for choosing to publish your research in The Cryosphere.

Pippa Whitehouse (Editor)

Minor editorial suggestions (line numbers refer to non-track-changed version of the article)

Line 16: suggest "…a transect of cores, *extending* from Nares Strait to *underneath* the 48 km ice tongue of Petermann Glacier, …"

Line 46: the statement about the 'emerging prominence of the northern arch' seems to be at odds with the previous sentence that documents a recent reduction in the number of days that Nares Strait is blocked each year. Please review the logic of these two sentences.

Line 52-53: the two halves of this sentence – before and after "in response to" – do not fit together very well. Please summarise the key points more clearly (they are covered very clearly in section 2)

Line 65: suggest "…a 4-year period with no/little landfast ice in Nares Strait, *and which was associated with the* earlier break-up of landfast in Petermann Fjord"

Line 97: "PGs" needs an apostrophe (PG's), but instead, I suggest "PGs" -> "the PG" (also line 62)

Lines 149-150: comparing with the second half of this sentence (lines 151-152), I suggest rephrasing the first part of the sentence to make it clearer that when there is mobile sea ice in Nares Strait, in spring/early summer, the ice edge is located within/at the mouth of Petermann Fjord

Line 314: suggest "…are presented *in terms of their depth within the core, to account for the fact that sediment ages are unconstrained in the bottom ca. 1.5 m of the core*."

Line 316: suggest "…, *which are* characteristic of…"

Line 329: results are generally reported working up the core, e.g. "between 154 cm and 70 cm" (line 331), i.e. forwards in time. However, some depths are reported from shallow to deep, e.g. "between 154 cm and 199 cm" (line 329). Please check for consistency throughout all the results sections.

Lines 394 and 395: should references to unit 1C actually be to unit 1A?

Line 519: suggest "…, *which is* characterized by…"

Line 551: suggest "…, *which were* associated with… "

Figure 6: please explain the blue/green +/- symbols and clarify the meaning of the labels 'IP$_{25}$' and 'Phyto. Marker' in the caption, e.g. when you mention "sea ice biomarker and primary productivity indicators". Apologies if I have missed this information elsewhere in the manuscript.

Line 587: suggest "…associated with variable HBI III fluxes *at Kane2b*, …"

Line 688: suggest "…this *coincides* with some driftwood landings…"

Line 694-695: I found this sentence a little confusing, would it be correct to say, "Georgiadis et al. (2020) interpret *low* HBI III as indicating ice loaded fresh surface waters *due* to mobile sea ice…"

Line 725: suggest "…*leads to* a regime…"

Line 744-746: consider re-ordering the ideas in this sentence, "Sea ice has the ability to influence the oceanic heat transport to Petermann Fjord by regulating the wind stress at the atmosphere/ocean interface in Nares Strait, which in turn modifies the circulation-driven inflow of AAW (reference)."

Line 758: (also mentioned by reviewer 1) I suggest "… is unlikely to have resulted in a prolonged…"

Line 767: including -> included ?

Line 775: "In light of the recent development" – please clarify what this refers to

Lines 791-792: "seasonal coastal melt" – not previously mentioned, please provide a reference or other supporting evidence